# Inferring Neural Signed Distance Functions by Overfitting on Single Noisy Point Clouds through Finetuning Data-Driven based Priors

Chao Chen[1]    Yu-Shen Liu[1]*    Zhizhong Han[2]
[1]School of Software, Tsinghua University, Beijing, China
[2]Department of Computer Science, Wayne State University, Detroit, USA
chenchao19@tsinghua.org.cn   liuyushen@mails.tsinghua.edu.cn
h312h@wayne.edu

## Abstract

It is important to estimate an accurate signed distance function (SDF) from a point cloud in many computer vision applications. The latest methods learn neural SDFs using either a data-driven based or an overfitting-based strategy. However, these two kinds of methods are with either poor generalization or slow convergence, which limits their capability under challenging scenarios like highly noisy point clouds. To resolve this issue, we propose a method to promote pros of both data-driven based and overfitting-based methods for better generalization, faster inference, and higher accuracy in learning neural SDFs. We introduce a novel statistical reasoning algorithm in local regions which is able to finetune data-driven based priors without signed distance supervision, clean point cloud, or point normals. This helps our method start with a good initialization, and converge to a minimum in a much faster way. Our numerical and visual comparisons with the state-of-the-art methods show our superiority over these methods in surface reconstruction and point cloud denoising on widely used shape and scene benchmarks. The code is available at https://github.com/chenchao15/LocalN2NM.

## 1 Introduction

It is an important task to estimate an implicit function from a point cloud in computer graphics, computer vision, and robotics. An implicit function, such as a signed distance function (SDF), describes a continuous 3D distance field to indicate distances to the nearest surfaces at arbitrary locations. Since point clouds are easy to obtain, they are widely used as an information source to estimate SDFs, particularly without using normals that are not available for most scenarios. The challenge for SDF estimation mainly comes from the difficulty of bridging the gap between the discreteness of point clouds and the continuity of implicit functions.

Recent methods [62, 64, 29, 14, 95, 80, 58, 74] overcome this challenge using either a data-driven based or an overfitting-based strategy. To map a point cloud to a signed distance field, the data-driven based methods [60, 27, 36, 45, 81, 79, 22, 42, 92, 83] rely on a prior learned with signed distance supervision from a large-scale dataset, while the overfitting-based methods [28, 1, 102, 2, 99, 4, 21, 50, 18, 88] do not need signed distance supervision and just use the point cloud to infer a signed distance field. However, both of the two kinds of methods have pros and cons. The data-driven based methods can do inference fast but suffers from the need of large-scale training samples and poor

---

*The corresponding author is Yu-Shen Liu. This work was supported by National Key R&D Program of China (2022YFC3800600), and the National Natural Science Foundation of China (62272263, 62072268), and in part by Tsinghua-Kuaishou Institute of Future Media Data.

38th Conference on Neural Information Processing Systems (NeurIPS 2024).

generalization to instances that are unseen during training. Although the overfitting-based methods have a better generalization ability and do not need the large-scale signed distance supervision, they usually require a much longer time to converge during inference. The cons of these two kinds of methods dramatically limit the performance of learning neural SDFs under challenging scenarios like highly noisy point clouds. Therefore, beyond pursuing higher accuracy of SDFs, how to balance the generalization ability and the convergence efficiency is also a significant issue.

To resolve this issue, we propose to learn an SDF from a single point cloud by finetuning data-driven based priors. Our key idea is to promote the advantages of both the data-driven based and the overfitting-based strategy to pursue better generalization, faster inference, and higher accuracy. Our method overfits a neural network on a single point cloud to estimate an SDF with a novel loss without using signed distance supervision, clean point, or point normals, where the neural network was pretrained as a data-driven based prior from large-scale signed distance supervision. With finetuning priors, our method can generalize better on unseen instances than the data-driven based methods, and also converge much more accurate SDFs in a much faster way than the overfitting-based methods. Moreover, our novel loss for finetuning the data-driven based prior can conduct a statistical reasoning in a local region which can recover more accurate and sharper underlying surface from noisy points. We report numerical and visual comparisons with the state-of-the-art methods and show our superiority over these methods in surface reconstruction and point cloud denoising on widely used shape and scene benchmarks. Our contributions are summarized below,

- We introduce a method which is capable of funetuning a data-driven based prior by minimizing an overfitting-based loss without signed distance supervision, leading to neural SDFs with better generalization, faster inference, and higher accuracy.

- The proposed overfitting-based loss can conduct a novel statistical reasoning in local regions, which improves the accuracy of neural SDFs inferred from noisy point clouds.

- Our method produces the state-of-the-art results in surface reconstruction and point cloud denoising on the widely used benchmarks.

## 2 Related Works

Learning implicit functions has achieved promising performance in various tasks [62, 64, 29, 14, 95, 80, 58, 74, 30, 31, 33]. We can learn neural implicit representations from different supervision including 3D supervision [61, 69, 59, 17], multi-view images [78, 44, 38, 101, 46, 94, 63, 41, 98, 97, 25, 86, 100, 89, 84, 85], and point clouds [92, 43, 60, 27]. We briefly review the existing methods related to point clouds below.

### 2.1 Data-driven based Methods

In 3D supervision, many techniques utilize a data-driven approach to learning priors, and then apply these learned priors to infer implicit models for unseen point clouds. Some strategies focus on acquiring global priors [60, 27, 36, 45, 81, 79, 22, 42] at the shape level, whereas others aim to boost the generalization of these priors by learning local priors [92, 83, 11, 37, 6, 51] at the component or patch level. These learned priors facilitate the marching cubes algorithm [47] to reconstruct surfaces from implicit fields. The effectiveness of these methods often rely on extensive datasets, but they may not generalize well when facing with unseen point clouds that significantly deviate in geometry from training samples.

### 2.2 Overfitting-based Methods

In an effort to enhance generalization, some methods concentrate on precisely fitting neural networks to single point clouds. These methods incorporate innovative constraints [28, 1, 102, 2, 99, 4, 21], utilize gradients [50, 18, 88], employ differentiable Poisson solvers [70], or apply specially tailored priors [51, 54] to learn either signed [50, 28, 1, 102, 2, 15, 56, 13] or unsigned distance functions [18, 104, 103]. Despite achieving significant advances, these approaches typically require clean point clouds to accurately determine distance or occupancy fields around the point clouds.

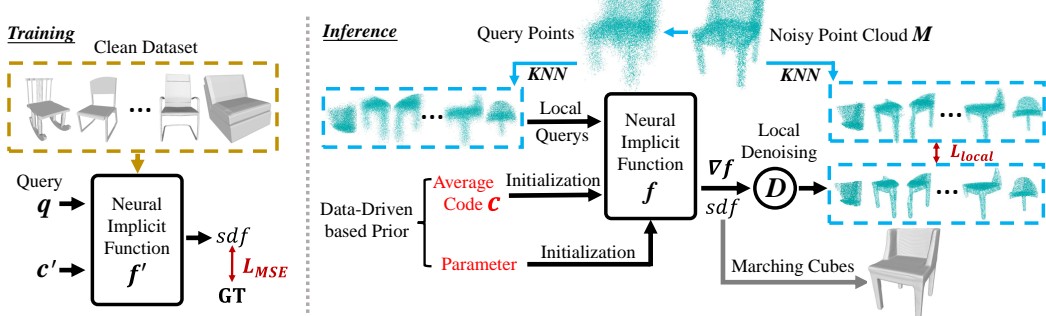

Figure 1: The overview of our method. We learn the data-driven based prior by learning a neural implicit function $f'$ with a condition $c'$ on a clean dataset. During inference, we employ a novel statistical reasoning algorithm to infer a neural SDF $f$ for a noisy point cloud $M$ with learned prior (average code and learned parameter).

## 2.3 Learning from Noisy Point Clouds

The key to accurately reconstructing surfaces on noisy point clouds is to minimize the effect of noise in inferring implicit functions. PointCleanNet [73] was developed to filter out noise from point clouds through a data-driven approach. GPDNet [72] incorporated graph convolution based on dynamically generated neighborhood graphs to enhance noise reduction. Some other methods leveraged point cloud convolution [6], alternating latent topology [90, 57], semi-supervised strategy [106, 19], dual and integrated latent [76], or neural kernel field [91, 35] to reduce noise from point clouds. On the unsupervised front, TotalDenoising [10] adopts principles similar to Noise2Noise [40], utilizing a spatial prior suitable for unordered point clouds. DiGS [3] employs a soft constraint for unoriented point clouds. Noise2NoiseMapping [52] leverage statistical reasoning among multiple noisy point clouds with specially designed losses. Some methods using downsample-upsample frameworks [48], gradient fields [49, 9, 16, 68, 65], convolution-free intrinsic occupancy network [67], intra-shape regularization [66], eikonal equation [96, 23], neural Galerkin [34] and neural splines [93] have been implemented to further diminish noise in point clouds. Our method falls in this category, but we aim to promote the advantages of both the data-driven based and the overfitting-based strategy to pursue better generalization, faster inference, and higher accuracy.

## 3 Method

**Overview.** We aim to infer a neural SDF $f$ from a single point cloud with noises $M$. Our method includes two stages shown in Fig. 1, one is to learn a prior $f'$ in a data-driven manner, the other is to infer a neural SDF $f$ on unseen noisy point cloud $M$. At the first stage, we learn a prior by training a neural SDF using ground truth signed distances of clean meshes indicated by embeddings $c'_j$. At the second stage, we finetune the learned prior $f'$ to infer a neural SDF $f$ of $M$ using our proposed local noise to noise mapping, where the embedding $c$ indicating $M$ is also learned. We can use the marching cubes algorithm [47] to extract the zero-level set of $f$ as the mesh surface of $M$.

**Neural Signed Distance Function.** We leverage an SDF $f$ to represent the geometry of a shape. An SDF $f$ is an implicit function that can predict a signed distance $s$ for an arbitrary location $q$, i.e., $s = f(q)$. The latest methods usually train a neural network to approximate an SDF from signed distance supervision or infer an SDF from 3D point clouds or multi-view images. A level set is an iso-surface formed by the points with the same signed distance value. For instance, zero-level set is a special level set, which is formed by points with a signed distance of 0. On the zero-level set, the gradient $\nabla f(q)$ of the SDF $f$ at an arbitrary location $q$ is also the surface normal at $q$.

**Data-driven Based Prior.** As shown in Fig. 1, we employ an auto-decoder similar to DeepSDF [69] for learning a prior $f'$ in a data-driven manner and inferring a neural SDF $f$ for single point clouds with noises, respectively. We employ a data-driven strategy to learn a prior $f'$ from clean meshes first. Specifically, we learn $f'$ with an embedding $c'_j$ as a condition of queries. For each shape, we sample queries $q$ around a shape represented by $c'_j$, and establish the signed distance supervision by recording the signed distance $s$ to the ground truth mesh. Thus, we learn the prior $f'$ by minimizing the prediction errors to the ground truth signed distances,

$$\min_{f',\{\boldsymbol{c}'_i\}} \sum_{i=1}^{I} \sum_{j=1}^{J} ||s_i^j - f'(q_j, \boldsymbol{c}'_i)||_2^2 + \alpha \sum_{i=1}^{I} ||\boldsymbol{c}'_i||_2^2, \tag{1}$$

where $\boldsymbol{c}'_i$ is a learnable condition for the $i$-th training shape, $q_j$ is the $j$-th query that is randomly sampled around the $i$-th shape, and $s_i^j$ is the ground truth signed distance. We also add a regularization term on the learned embeddings $\boldsymbol{c}'_i$, and $\alpha$ is the balance weight.

**Signed Distance Inference.** With the learned prior $f'$, we infer a neural SDF $f$ for a single point cloud with noises $M$. We do not require ground truth signed distances, clean point clouds, or even point normal during the inference of $f$. Specifically, we infer $f$ by finetuning parameters of $f'$ with a learnable embedding $\boldsymbol{c}$ indicating the single point cloud with noises. The finetuning relies on a novel statistical reasoning algorithm on local regions.

The advantage of our method lies in the capability of conducting the statistical reasoning in local regions. Comparing to the global reasoning method [52], our method is able to not only infer more accurate geometry but also significantly improve the efficiency. Our method starts from randomly sampling a local region $m_n$ on the shape $M$. We randomly select one point on $M$ as the center of $m_n$, and set up its $K$ nearest noisy points as a local region $m_n$. Then, we randomly sample $U$ queries $\{\bar{q}_u\}_{u=1}^{U}$ around $m_n$, and also randomly select $U$ noisy points $\{p_v\}_{v=1}^{U}$ out of $m_n$ for statistically reasoning the surface in each iteration.

Our key idea of inferring a neural SDF $f$ is to estimate a mean zero-level set that is consistent to all points in the local region $m_n$. To this end, we use the $U$ sampled queries $\{\bar{q}_u\}$ to represent the zero-level set in this area using $f$, and minimize the distances of the $U$ noisy points $\{p_v\}$ to the zero-level set in each iteration. Statistically, the expectation of the zero-level set should have the minimum distance to all the noisy point splitting in region $m_n$.

Specifically, we first project the $U$ sampled queries $\{\bar{q}_u\}$ onto the zero-level set of $f$ using a differentiable pulling operation [50]. For each query $\bar{q}_u$, its projection on the zero-level set is,

$$\bar{q}'_u = \bar{q}_u - s * \nabla f(\bar{q}_u, \boldsymbol{c})/|\nabla f(\bar{q}_u, \boldsymbol{c})|, \tag{2}$$

where $\bar{q}'_u$ is the projection of $\bar{q}_u$ on the zero-level set, $s = f(\bar{q}_u, \boldsymbol{c})$, $\nabla f(\bar{q}_u, \boldsymbol{c})$ is the gradient of $f$ at the location $\bar{q}_u$, and $\boldsymbol{c}$ is the learnable embedding that represents the noisy point cloud $M$.

With the pulling operation, we can use projections $\{\bar{q}'_u\}$ of queries $\{\bar{q}_u\}$ to approximate the zero-level set in region $m_n$. With a coarse zero-level set estimation, we expect this zero-level set can be consistent to various subsets of noises $\{p_v\}$ sampled from $m_n$. Thus, we minimize the errors between the $\{\bar{q}'_u\}_{u=1}^{U}$ and a subset of points $\{p_v\}_{v=1}^{U}$ on area $m_n$ in each optimization iteration,

$$\min_{f,\boldsymbol{c}} \mathbb{E}_{m_n \sim M, \bar{q}_u \sim m_n, p_v \sim m_n} EMD(\{\bar{q}'_u\}, \{p_v\}) + \beta ||\boldsymbol{c}||_2^2, \tag{3}$$

where we learn $f$ through finetuning the prior $f'$ and learning the embedding $\boldsymbol{c}$ representing the noisy point cloud $M$. The expectation is over the local regions $m_n$ that randomly sampled from the noisy point cloud $M$, and the subset patch $p_v$ randomly sampled from each $m_n$. We follow the method [52] to use the EMD to evaluate the distance between the two sets of points, which leads the neural SDF $f$ to converge on the specific noisy point cloud $M$.

**Initialization.** The network architecture of $f$ is the same to the one of prior $f'$. We learn $f$ with the parameters of $f'$ as the initialization, representing the prior that we learned. For the embedding $\boldsymbol{c}$ that represents $M$, we initialize $\boldsymbol{c}$ as the center of the embedding space learned by the prior $f'$ in Eq. 1, i.e., $\boldsymbol{c} = 1/I \sum_{i=1}^{I} \boldsymbol{c}'_i$. This initialization is important for the accuracy and efficiency of learning $f$ for single noisy point cloud $M$. This finetuning of parameters of $f'$ also shows advantages over the auto-decoding [69] in terms of generalization and efficiency. We will justify these advantages in our experiments.

**Implementation Details.** We randomly select one point from noisy point cloud $M$ as a center, and select its $K = 1000$ nearest points to form a local region $m_n$. We also randomly sample $U = 1000$ queries around the $K$ noisy points for statistically reasoning. Specifically, we adopt a method

| Metrics | PSR [39] | PSG [24] | R2N2 [20] | COcc [71] | SAP [70] | OCNN [87] | IMLS [45] | POCO [7] | ALTO [90] | N2NM [52] | Ours |
|---|---|---|---|---|---|---|---|---|---|---|---|
| $CD_{L1}$ | 0.299 | 0.147 | 0.173 | 0.044 | 0.034 | 0.067 | 0.031 | 0.030 | 0.028 | 0.026 | **0.023** |
| NC | 0.772 | - | 0.715 | 0.938 | 0.944 | 0.932 | 0.944 | 0.950 | 0.955 | 0.962 | **0.973** |
| F-Score | 0.612 | 0.259 | 0.400 | 0.942 | 0.975 | 0.800 | 0.983 | 0.984 | 0.985 | 0.991 | **0.992** |

Table 1: Numerical Comparisons on ShapeNet dataset in terms of $CD_{L1} \times 10$, NC and F-Score.

introduced by NeuralPull [50] to sample queries around each one of the $K$ noisy points. We use a Gaussian distribution centered at each point and set the standard deviation as the distance to the 51th nearest neighbor in the point cloud. We run the marching cubes for surface reconstruction at a resolution of 256 for shapes, and 512 for large-scale scenes.

The length of the embedding $c$ or $c'$ is set to 256. We use Adam optimizer for learning a neural implicit network, which is an auto-decoder similar to DeepSDF [69]. For training, we use an initial embedding learning rate of 0.0005 for updating embeddings and an auto-decoder learning rate of 0.001 for optimizing the prior network. Both learning rates are decreased by 0.5 for every 500 epochs. We train the prior network $f'$ for 2000 epochs. For inference, we finetune the network $f'$ for each noisy point cloud in 4000 iterations with a learning rate of 0.0001.

## 4  Experiments and Analysis

We compare our method with the latest methods in terms of numerical and visual results on synthetic point clouds and real scans in surface reconstruction.

**Datasets and Metric.**  We use eight datasets including shapes and scenes in the evaluations. For shapes, we conduct experiments under five datasets including ShapeNet [12], ABC [22], FA-MOUS [22], Surface Reconstruction Benchmark (SRB) [92] and D-FAUST [5]. For scenes, we conduct experiments under three real scan datasets including 3D Scene [105], KITTI [26], Paris-rue-Madame [75], and nuScenes [8]. We leverage L1 Chamfer Distance ($CD_{L1}$), L2 Chamfer Distance ($CD_{L2}$) to evaluate the error between the reconstructed surface and ground truth. We also use Normal Consistency (NC) [59] and F-Score [82] with a threshold of 1% to evaluate the normal accuracy of the reconstructed surface. In the ablation study, we also report time consumption to highlight the superiority of our data-driven based prior. For KITTI and Paris-rue-Madame datasets, due to their lack of ground truth meshes, we only report visual comparisons.

### 4.1  Surface Reconstruction for Shapes

**Evaluation on ShapeNet.** We first report our results on shapes from ShapeNet. We report evaluations by comparing our method with the latest prior-based and overfitting-based methods in Tab 1. For prior-based methods, we compare our method with PSG [24], R2N2 [20], COcc [71], OCNN [87], IMLS [45], POCO [7], and ALTO [90]. All of these methods are pretrained to learn priors using shapes with noises in training set of ShapeNet. We also follow these meth-

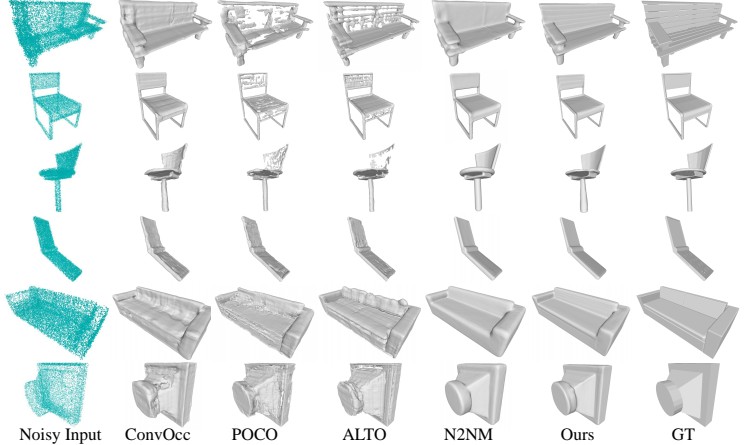

Noisy Input   ConvOcc   POCO   ALTO   N2NM   Ours   GT

Figure 2: Comparison in surface reconstruction on ShapeNet. More visual results are provided in the appendix.

ods to use the same set of training shapes to learn our prior. For overfitting-based methods, we compare our method with PSR [39], SAP [70], and N2NM [52]. These methods did not need to learn a prior, and have the ability of inferring neural implicit functions on each shape in the testing set. We also follow these methods and report our results by finetuning our prior through overfitting on each testing shape. All the shapes for testing are corrupted with noises with a variance of 0.005.

The comparisons in Tab. 1 indicate that our method can infer much more accurate neural implicit functions than the prior-based methods. The improvement comes from the ability of conducting test time optimization with the learned prior and inferring signed distances using the local noise to noise mapping. Moreover, our

| Metrics | SAP [70] | N2NM [52] | Ours |
|---------|----------|-----------|------|
| Time    | 14 min   | 46 min    | **5 min** |

Table 2: Time consumption on ShapeNet dataset with overfitting-based methods.

local statistical reasoning not only achieves better ability of recovering geometry from noisy points than overfitting-based methods but also significantly reduces the time complexity during the test time overfitting procedure with our prior. Different from prior-based methods, our ability of conducting test-time optimization with our local statistical reasoning loss can significantly improve the generalization ability on unseen shapes. Tab. 2 shows that our method can infer neural implicit functions on single shapes much faster than the overfitting-based methods. We also demonstrate our advantages in visual comparisons in Fig. 2.

**Evaluation on ABC.** We also report our evaluations on ABC dataset in Tab. 3. We learn priors from shapes in training set, and finetune this prior for each single shape in the testing set. The numerical comparisons are conducted on the testing set of ABC dataset released by P2S [22]. It includes two versions with different

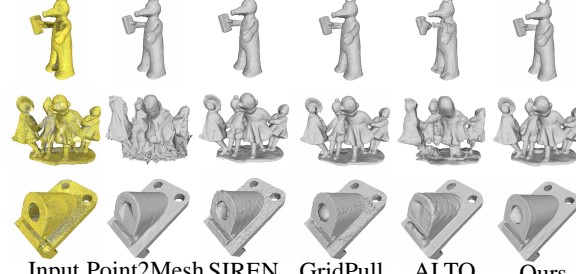

Figure 3: Comparison in surface reconstruction on ABC. More visual results are provided in the appendix.

noise levels. Similarly, we also report comparisons with prior-based methods and overfitting-based methods. With our local noise to noise mapping, we achieve the best performance over all baselines. Compared to prior-based methods, such as P2S [22], COcc [71], and POCO [7], our loss can infer more accurate geometry during the test time overfitting procedure. Also, the ability of finetuning the prior can also provide a coarse estimation and a good start for inferring neural implicit from single noisy points. Besides the accuracy, we also observe improvements on efficiency. Fig. 3 demonstrates the improvements over the baselines in terms of surface completeness and edge sharpness.

**Evaluation on SRB.** We report previous experiments using man-made objects in ShapeNet and ABC dataset, We also report our results on real scans on SRB dataset [92]. Since there is no training samples on SRB, we use the prior learned from the ShapeNet as the prior for real scans. Although the shapes in ShapeNet are not similar to shapes in SRB, we found the prior can also work well with the scans on SRB. Different from the man-made objects, real scans have unknown noises. We report the

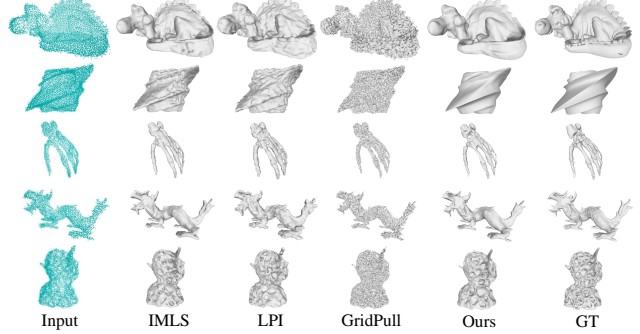

Figure 4: Comparison in surface reconstruction on SRB. More visual results are provided in the appendix.

evaluations with the prior-based and overfitting-based methods in Tab. 4 and Fig. 4. The comparisons show that our method achieves the best performance in implicit surface reconstruction. Under the same experimental settings, our method can infer more accurate geometry details with our local noise to noise mapping.

**Evaluation on FAMOUS.** We report evaluations on more complex shapes on FAMOUS dataset. Similar to SRB, we also use the prior learned from ShapeNet. We evaluate the performance on two kinds of noises in Tab. 5. We can see that our method can recover more geometry details and achieve higher accuracy and smoother surfaces. We also report visual comparisons in Fig. 5, which also highlights our improvements in

Figure 5: Comparison in surface reconstruction on FAMOUS. More visual results are provided in the appendix.

| Dataset | PSR [39] | P2S [22] | COcc [71] | NP [50] | IMLS [45] | PCP [55] | POCO [7] | OnSurf [53] | N2NM [52] | Ours |
|---|---|---|---|---|---|---|---|---|---|---|
| ABC var | 3.29 | 2.14 | 0.89 | 0.72 | 0.57 | 0.49 | 2.01 | 3.52 | 0.113 | **0.096** |
| ABC max | 3.89 | 2.76 | 1.45 | 1.24 | 0.68 | 0.57 | 2.50 | 4.30 | 0.139 | **0.113** |

Table 3: Numerical Comparisons on ABC dataset in terms of $CD_{L2} \times 100$.

| Metrics | IGR [28] | Point2Mesh [32] | PSR [39] | SIREN [77] | GP [16] | ALTO [90] | Steik [96] | SAP [70] | NKSR [35] | N2NM [52] | Ours |
|---|---|---|---|---|---|---|---|---|---|---|---|
| $CD_{L1}$ | 0.178 | 0.116 | 0.232 | 0.123 | 0.086 | 0.089 | 0.079 | 0.076 | 0.069 | 0.067 | **0.055** |
| F-Score | 0.755 | 0.648 | 0.735 | 0.677 | 0.766 | 0.772 | 0.822 | 0.830 | 0.829 | 0.835 | **0.860** |

Table 4: Numerical Comparisons on SRB dataset in terms of $CD_{L1} \times 10$ and F-Score.

terms of accuracy, smoothness, completeness, and recovered sharp edges.

**Evaluation on D-FAUST.** Finally, we report our results on non-rigid shapes, i.e., humans. Different from rigid shapes in the previous experiments, humans are with more complex poses. We learn a prior from the training set, and finetuning the prior on unseen humans with different poses. We mainly compare our method with overfitting-based methods in Tab. 6. We can see that our method achieves the best performance in CD, F-Score, and comparable performance to N2NM [52] but with faster inference speed. We further show the visual comparison in Fig. 6. We can see that our method can recover more accurate geometry and poses.

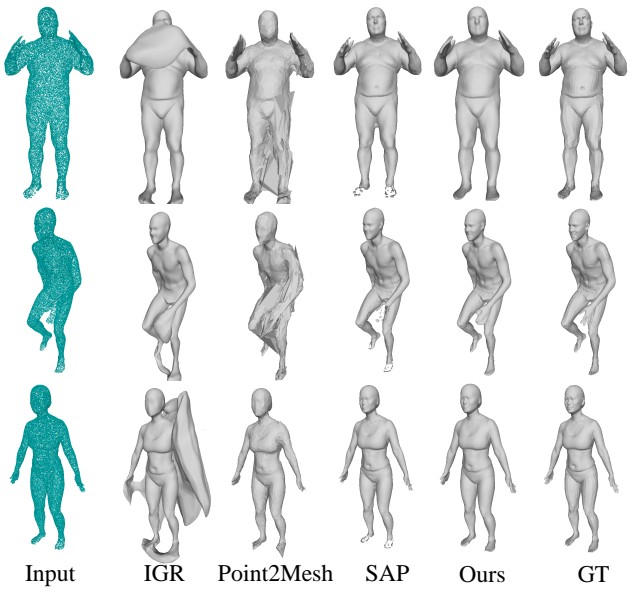

Input    IGR    Point2Mesh    SAP    Ours    GT

Figure 6: Comparison in surface reconstruction on D-FAUST. More visual results are provided in the appendix.

## 4.2 Surface Reconstruction for Scenes

Since we have a limited number of scenes for training, we use the prior learned from ShapeNet as the pretrained prior in our experiments on scenes. Specifically, we conduct experiments on four different scene datasets: 3D Scene [105], KITTI [26], Paris-rue-Madame [75] and nuScenes [8], where the results on nuScenes are reported in the appendix.

**Evaluation on 3D Scene.** We further evaluate our method in surface reconstruction for scenes in 3D Scene [105]. We follow previous methods LIG [37] to randomly sample 1000 points per $m^2$. We compare our method with the latest methods including COcc [71] and LIG [37], DeepLS [11], NeuralPull (NP) [50] and Noise2NoiseMapping (N2NM) [52]. For prior-based methods COcc [71] and LIG [37], we leverage their released pretrained models to produce the results, and we also provide them with the ground truth point normals. For overfitting-based methods DeepLS [11], NP [50] and N2NM [52], we overfit them to produce results with the same noisy point clouds. We follow LIG [37] to report $CD_{L1}$, $CD_{L2}$ and NC for evaluation. We report the comparisons in Tab. 7. The results demonstrate that our method outperforms both kinds of methods with learned priors such as LIG [37] and overfitting-based N2NM [52]. The visual comparisons in Fig. 7 show that our method can reveal more geometry details on real scans, which justifies our capability of handling noise in point clouds.

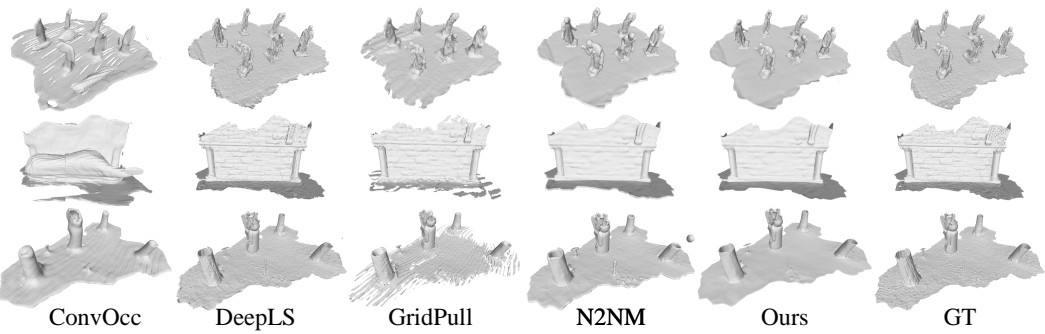

ConvOcc    DeepLS    GridPull    **N2NM**    Ours    GT

Figure 7: Comparison in surface reconstruction on 3D Scene.

| Dataset | PSR [39] | NP [50] | IMLS [45] | LPI [15] | PCP [55] | POCO [7] | OnSurf [53] | GP [16] | N2NM [52] | Ours |
|---|---|---|---|---|---|---|---|---|---|---|
| F-var | 1.80 | 0.28 | 0.80 | 0.19 | 0.07 | 1.50 | 0.59 | 0.13 | 0.033 | **0.029** |
| F-max | 3.41 | 0.31 | 0.39 | 0.26 | 0.30 | 2.75 | 3.64 | 0.21 | 0.117 | **0.105** |

Table 5: Numerical Comparisons on FAMOUS dataset in terms of $CD_{L2} \times 100$.

| Metrics | IGR [28] | Point2Mesh [32] | PSR [39] | SAP [70] | N2NM [52] | Ours |
|---|---|---|---|---|---|---|
| $CD_{L1} \times 10$ | 0.235 | 0.071 | 0.044 | 0.043 | 0.037 | **0.034** |
| F-Score | 0.805 | 0.855 | 0.966 | 0.966 | 0.966 | **0.973** |
| NC | 0.911 | 0.905 | 0.965 | 0.959 | **0.970** | 0.968 |

Table 6: Accuracy of reconstruction on D-FAUST dataset in terms of $CD_{L1}$, NC and F-Score.

**Evaluation on KITTI.** Following GridPull [16], we further evaluate our method on KITTI [26] odometry dataset (Sequence 00, frame 3000 to 4000), which contains about 13.8 million points, which are split into 15 chunks. We reconstruct each of them and concatenate them together for visualization. We compare our method with the latest methods SAP [70] and GridPull [16]. As shown in Fig. 8, our method is robust to noise in real scans, successfully generalizes to large-scale scenes, and achieves visual-appealing reconstructions with more details.

**Evaluation on Paris-rue-Madame.** Following N2NM [52], we further evaluate our method on Paris-rue-Madame [75], which contains much noises. We split the 10 million points into 50 chunks each of which is used to learn a neural implicit function. We compare our method with LIG [37] and N2NM [52]. For LIG [37], we produce the results for each chunk with released pretrained models. For N2NM [52], we overfit on all chunks until convergence. As shown in Fig. 9, we achieve better performance over LIG [37] and N2NM [52] in large-scale surface reconstruction, which highlight our advantages in reconstructing complete and detailed surfaces from noisy scene point clouds.

## 4.3 Ablation Studies

We conduct ablation studies on the ABC dataset [22] to justify each module of our method.

**Embedding Size.** We evaluate our performance on different sizes of embedding $c$. We try several sizes $\{128, 256, 512\}$ to infer the signed distance functions from a noisy point cloud. The numerical comparison in Tab. 8 shows that the optimal result is obtained with a size of 256. Deviations from this value, either longer or shorter dimensions, leads to worse results with the current number of training samples.

| Metric | 128 | 256 | 512 |
|---|---|---|---|
| $CD_{L2} \times 100$ | 0.102 | **0.096** | 0.114 |

Table 8: Effect of the embedding size.

**Prior.** We conduct experiments to explore the importance of data-driven based prior. We first replace our learned embedding $c$ and parameter with randomly initialized embedding and parameter, or only replace $c$ with randomly initialized embedding. As shown in Tab. 9, The degenerated result of "Without Prior" and "Without Embed" indicates that directly inferring implicit functions without our prior or learned embedding makes it difficult to accurately learn the surfaces of the noisy point clouds, and also slows the convergence. Then we fix the learned parameters and only optimize the embedding $c$, similar to auto-decoding. The results also get worse, as shown in "Fixed Param".

| Metric | Without Prior | Without Embed | Fixed Param | With Prior |
|---|---|---|---|---|
| $CD_{L2} \times 100$ | 0.108 | 0.103 | 0.144 | **0.096** |
| Time | 1h | 12min | 30min | **8 min** |

Table 9: Effect of the prior.

**Local Region Splitting.** We further validate the effectiveness of local region splitting strategies. We employ three different splitting strategies in Tab. 10. We first split the whole space where the noisy point cloud is located uniformly into multiple voxel blocks, as shown by the result of "Voxel". The severely degenerated results indicate that this splitting strategy is even worse than the global method N2NM [52], as it results in many empty voxel blocks. Then we randomly select a point from the noisy point cloud as a center to sample all points within a radius of 0.1 as a local region. The result of "Sphere (Fixed Size)" slightly degenerates due to some of the spheres containing too few points. In contrast, our splitting strategy, as shown by the result of "Sphere (KNN)", ensures that each local region has enough points to help achieve superior performance.

| Metric | Voxel | Sphere (Fixed Size) | Sphere (KNN) |
|---|---|---|---|
| $CD_{L2} \times 100$ | 0.314 | 0.101 | **0.096** |

Table 10: Effect of splitting strategies.

| Metrics | COcc [71] | LIG [37] | DeepLS[11] | NP [50] | N2NM [52] | Ours |
|---|---|---|---|---|---|---|
| $CD_{L2} \times 1000$ | 14.10 | 6.190 | 1.607 | 2.115 | 0.507 | **0.389** |
| $CD_{L1}$ | 0.052 | 0.048 | 0.025 | 0.034 | 0.019 | **0.016** |
| NC | 0.908 | 0.849 | 0.915 | 0.900 | 0.929 | **0.942** |

Table 7: Numerical Comparisons on 3D Scene dataset in terms of $CD_{L1}$, $CD_{L2}$ and NC. Detailed comparisons for each scene are provided in the appendix.

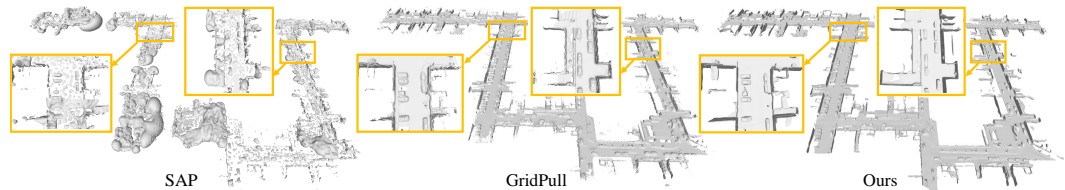

SAP                          GridPull                      Ours

Figure 8: Comparison in surface reconstruction on KITTI.

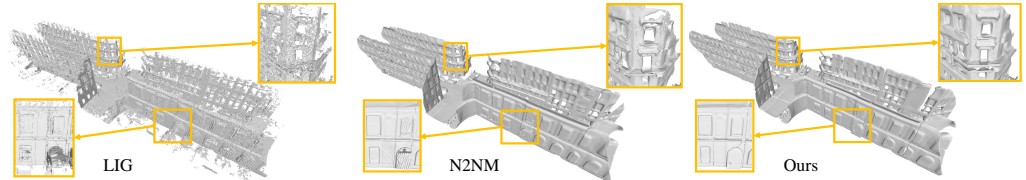

LIG                         N2NM                        Ours

Figure 9: Comparison in surface reconstruction on Paris-rue-Madame.

**Global and Local.** With our learned prior, we compare our performance in global and local mappings with finetuning the priors. We report results obtained with the local noise to noise mapping or the global one during the finetuning. As shown in Tab. 11, the numerical comparison shows that the global mapping struggles to infer local details from noisy point clouds. Moreover, our local prior also converges faster than the global statistical reasoning.

| Metric | Global | Local |
|---|---|---|
| $CD_{L2} \times 100$ | 0.106 | **0.096** |
| Time | 21 min | **8 min** |

Table 11: Effect of local mapping.

**Local Region Size.** We further validate the effectiveness of local region sizes (points number in a local region) in Tab. 12. We use different local region sizes including $\{500, 1000, 3000, 5000\}$. The results show that 1000 is the best.

| Metric | 500 | 1000 | 3000 | 5000 |
|---|---|---|---|---|
| $CD_{L2} \times 100$ | 0.102 | **0.096** | 0.111 | 0.114 |

Table 12: Effect of local region size.

**SDF initialization.** We further validate the effectiveness of different SDF initializations in Tab. 13 and Fig. 10, including random initialization, geometry initialization [1], initialization to a simple square shape, and ours.

| Metric | Random | Square | Sphere (SAL) | Ours |
|---|---|---|---|---|
| Time | 8.3min | 7.1min | 5.5min | **5.0min** |

Table 13: The effect of SDF initialization.

We can see our prior can reconstruct more accurate surfaces from single noisy point clouds in much shorter time than any other initializations.

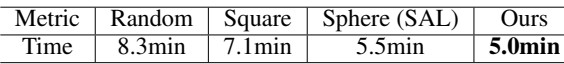

Random     Square     Sphere     Ours     GT

Figure 10: Comparison with different SDF initializations.

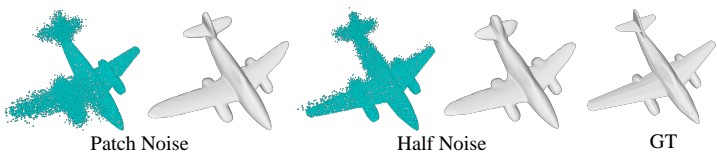

Patch Noise           Half Noise        GT

Figure 11: Visual results with nonuniform noises.

**Noise Type.** We report our performance with various noise types, i.e., impulse noise, quantization noise, Laplacian noise, and Gaussian noise. Visual comparison in Fig. 12 justifies that we can also handle other types of noise quite well. Moreover, we also tried more challenging cases with nonuniform noises which do not have a zero expectation across a shape, like a shape with only a half of points having noises or a shape with several patches having noises. The result in Fig. 11 shows that our method can also handle nonuniform noises well.

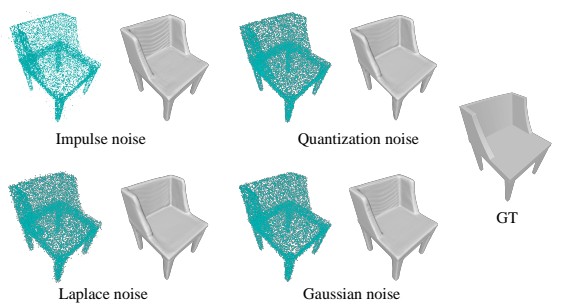

Impulse noise          Quantization noise

Laplace noise          Gaussian noise     GT

Figure 12: Visual results with different noise types.

**Noise Level.** We report our performance on point clouds with different levels of noise. As shown in Tab. 14, the noise levels of middle and max come from the ABC dataset [22]. The middle indicates noises

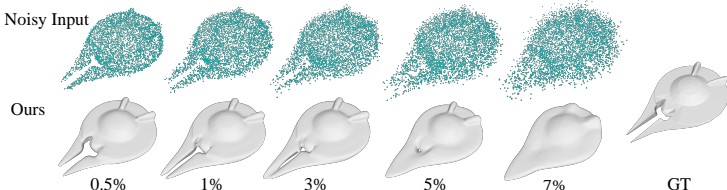

Figure 13: Visual comparison with different noise levels.

with a variance of $0.01L$, where $L$ is the longest edge of the bounding box. The max indicates noises with a variance of $0.05L$. Our extreme noise comes with a variance of $0.07L$.

The $CD_{L2}$ comparison shows that our results slightly degenerate with max and extreme noise, but still outperform N2NM [52]. The visual results in Fig. 13 indicates that our method is more robust to noises even when the noise variance is as large as 7%.

| Method | Middle | Max | Extreme |
|---|---|---|---|
| N2NM [52] | 0.113 | 0.139 | 0.156 |
| Ours | **0.096** | **0.113** | **0.125** |

Table 14: Effect of noise level.

**Sparsity.** We report the effect of the sparsity of noisy point clouds. We downsample the noisy point clouds to 25% and 50% of their original size to validate the impact of sparsity. The $CD_{L2}$ results in Tab. 15 and visual comparisons in Fig. 14 indicate that our method can handle sparsity in noisy point

| Method | 25% | 50% | 100% |
|---|---|---|---|
| N2NM [52] | 0.154 | 0.133 | 0.113 |
| Ours | **0.121** | **0.107** | **0.096** |

Table 15: Effect of sparsity.

clouds better than N2NM [52]. Since our data-driven based prior can help to learn a more complete surface and reduce the impacts brought by the sparsity.

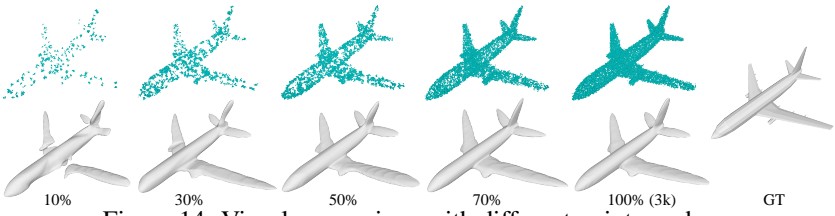

Figure 14: Visual comparison with different point numbers.

**Time Consumption.** Since our method can handle sparsity and require less time as the point number decreases, we conduct an experiment with downsampled noisy points in Tab. 16. Fig. 14 indicates that we can work well on much fewer points, and also provide an alternative of improving efficiency.

| Metric | 10% | 30% | 50% | 70% | 100% (3k) |
|---|---|---|---|---|---|
| Time | 3.1min | 3.6min | 4.0min | 4.5min | 5.0min |

Table 16: The comparison of time consumption with different point numbers.

**Optimization.** We visualize the optimization process in Fig. 15. We reconstruct meshes using the neural SDF $f$ learned in different iterations. We

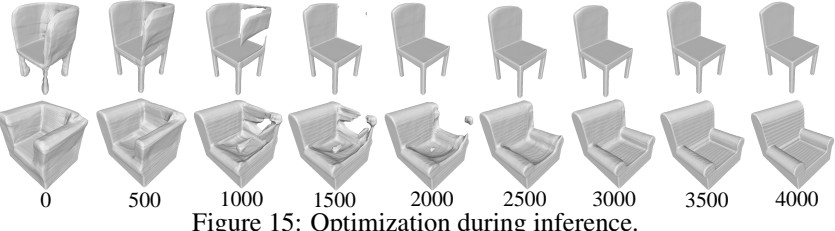

Figure 15: Optimization during inference.

see that the shape is updated progressively to the ground truth shapes.

## 5 Conclusion

We propose a method to resolve the key problem in inferring SDFs from a single noisy point cloud. Our method can effectively use a data-driven based prior as an initialization, and infer a neural SDF by overfitting on a single noisy point cloud. The novel statistical reasoning successfully infers an accurate and smooth signed distance field around the single noisy point cloud with the data-driven based prior. By finetuning data-driven based priors with statistical reasoning, our method significantly improves the robustness, the scalability, the efficiency, and the accuracy in inferring SDFs from single point clouds. Our experimental results and ablations studies show our superiority and justify the effectiveness of the proposed modules.

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

# A  Appendix

## A.1  Limitations

Our method is still limited to too large noises. For noises that corrupted shapes too much, our method still produces bad results. One direction for our future work is to improve our prior, so that we could have a better sense of a shape even under large noises.

## A.2  Detailed Comparisons on 3D Scene

We detail our evaluations on each scene in 3D scene dataset in Tab. 17. The comparisons highlight our advantages in each scene.

| Name | Metrics | COcc [71] | LIG [37] | DeepLS[11] | NP [50] | N2NM [52] | Ours |
|------|---------|-----------|----------|------------|---------|-----------|------|
| Burghers | $CD_{L2} \times 1000$ | 27.46 | 3.055 | **0.401** | 1.204 | 0.504 | 0.429 |
| | $CD_{L1}$ | 0.079 | 0.045 | 0.017 | 0.031 | 0.020 | **0.016** |
| | NC | 0.907 | 0.835 | 0.920 | 0.905 | 0.925 | **0.939** |
| Lounge | $CD_{L2} \times 1000$ | 9.540 | 9.672 | 6.103 | 1.079 | 0.602 | **0.333** |
| | $CD_{L1}$ | 0.046 | 0.056 | 0.053 | 0.019 | 0.016 | **0.014** |
| | NC | 0.894 | 0.833 | 0.848 | 0.910 | 0.923 | **0.935** |
| Copyroom | $CD_{L2} \times 1000$ | 10.97 | 3.610 | 0.609 | 5.795 | 0.442 | **0.389** |
| | $CD_{L1}$ | 0.045 | 0.036 | 0.021 | 0.036 | **0.016** | 0.016 |
| | NC | 0.892 | 0.810 | 0.901 | 0.862 | 0.903 | **0.916** |
| Stonewall | $CD_{L2} \times 1000$ | 20.46 | 5.032 | 0.320 | 0.983 | 0.330 | **0.313** |
| | $CD_{L1}$ | 0.069 | 0.042 | 0.015 | 0.029 | 0.020 | **0.015** |
| | NC | 0.905 | 0.879 | 0.954 | 0.930 | 0.951 | **0.961** |
| Totepole | $CD_{L2} \times 1000$ | 2.054 | 9.580 | 0.601 | 1.513 | 0.657 | **0.482** |
| | $CD_{L1}$ | 0.021 | 0.062 | **0.017** | 0.054 | 0.023 | 0.020 |
| | NC | 0.943 | 0.887 | 0.950 | 0.893 | 0.945 | **0.957** |

Table 17: Numerical Comparisons on 3D Scene dataset in terms of $CD_{L1}$, $CD_{L2}$ and NC.

## A.3  More Results

We visualize more surface reconstruction results under ShapeNet [12], ABC [22], Surface Reconstruction Benchmark (SRB) [92], FAMOUS [22], D-FAUST [5] and nuScenes [8] in Fig. 16, Fig. 17, Fig. 18, Fig. 19, Fig. 20 and Fig. 21.

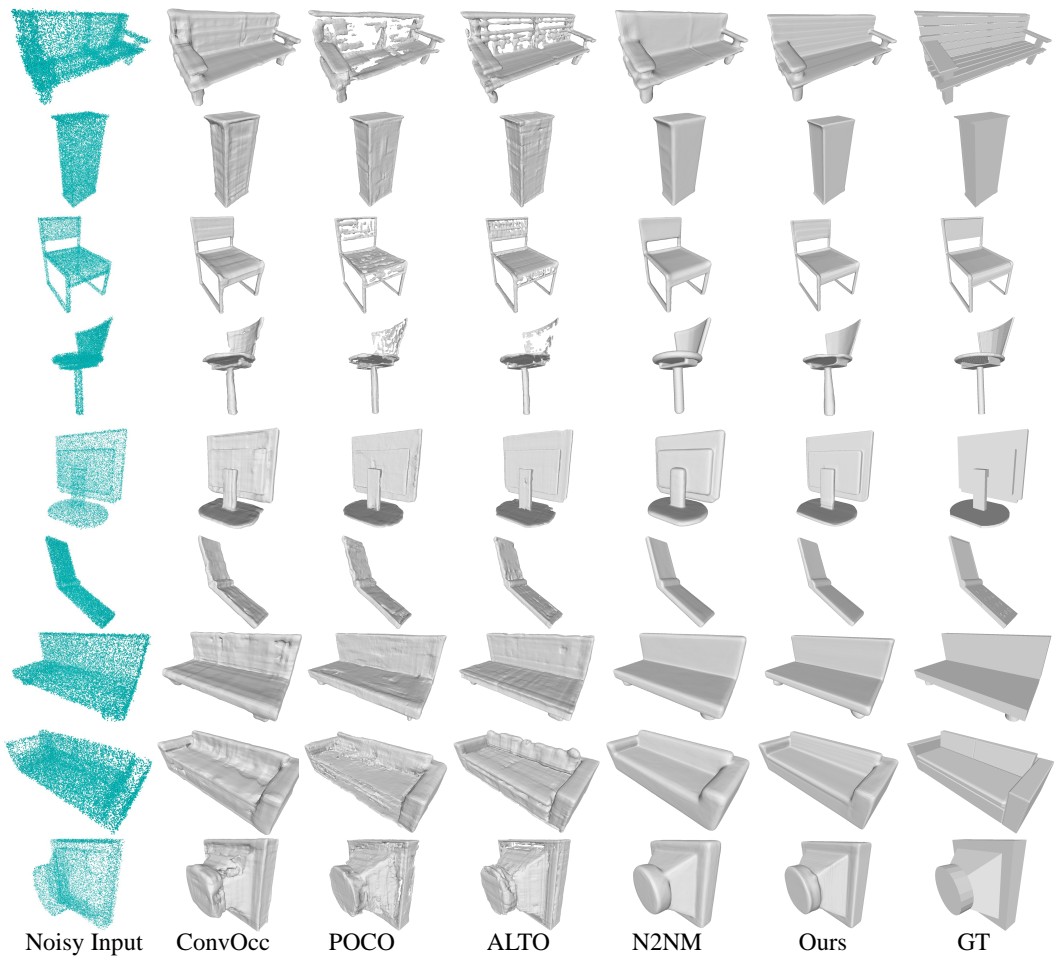

Figure 16: Comparison in surface reconstruction on ShapeNet.

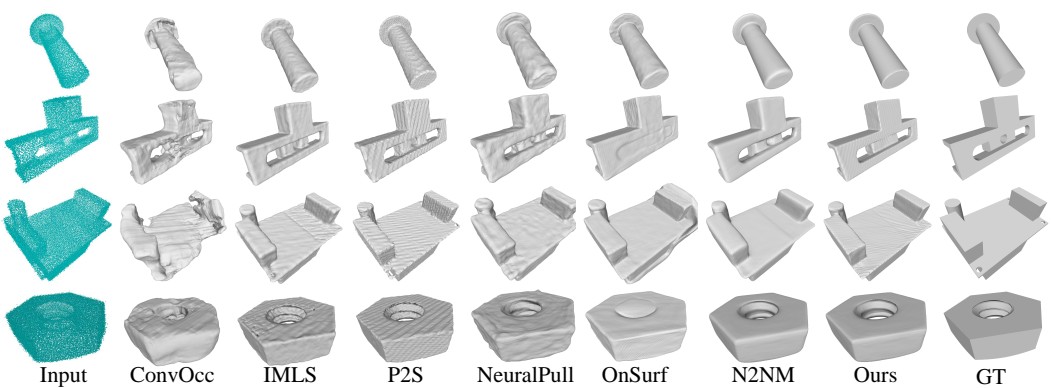

Figure 17: Comparison in surface reconstruction on ABC.

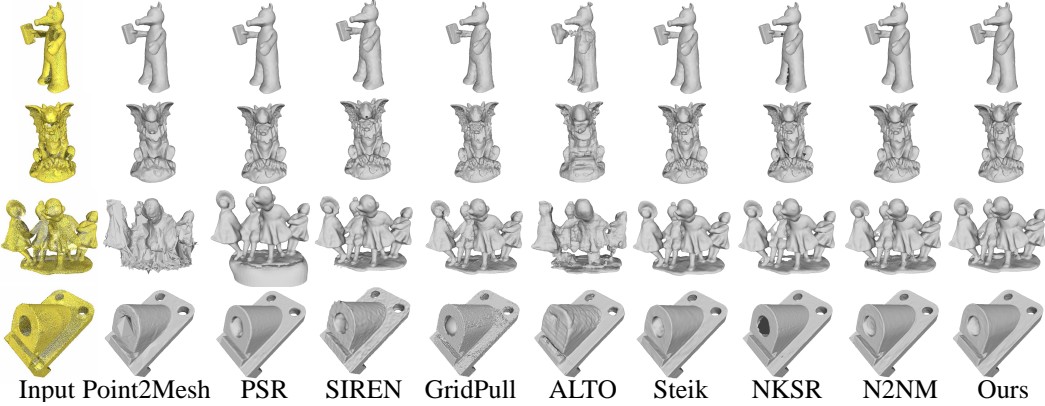

Input  Point2Mesh  PSR  SIREN  GridPull  ALTO  Steik  NKSR  N2NM  Ours

Figure 18: Comparison in surface reconstruction on SRB.

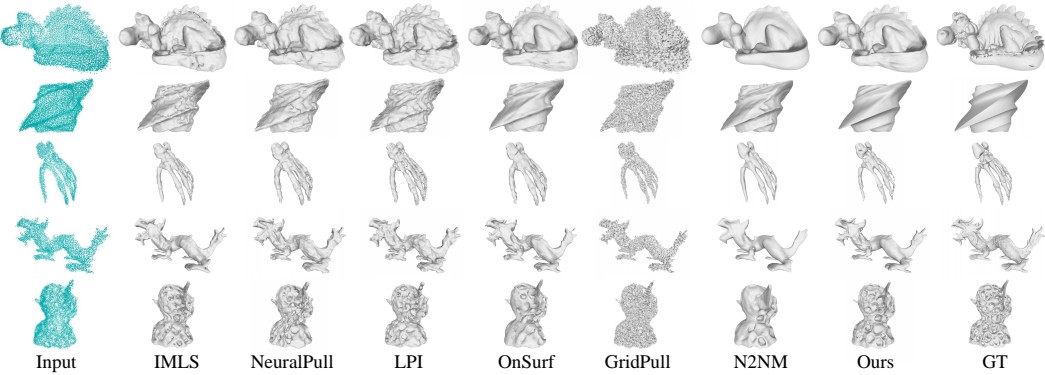

Input  IMLS  NeuralPull  LPI  OnSurf  GridPull  N2NM  Ours  GT

Figure 19: Comparison in surface reconstruction on FAMOUS.

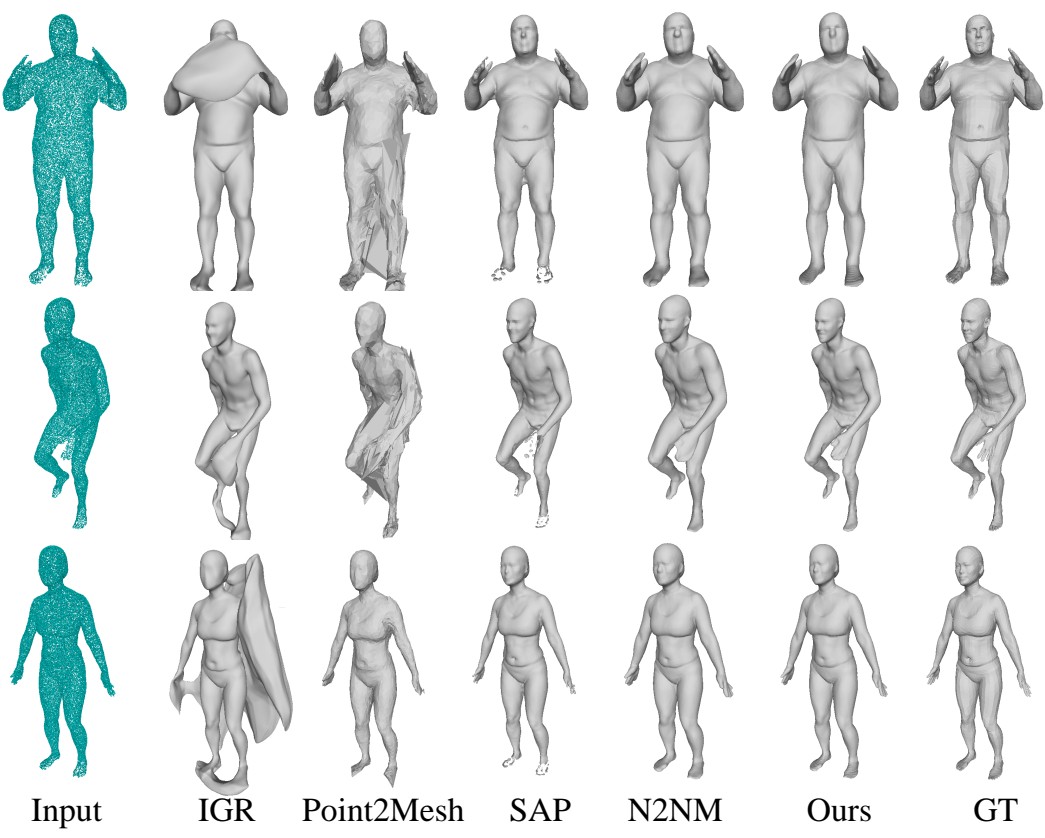

| Input | IGR | Point2Mesh | SAP | N2NM | Ours | GT |

Figure 20: Comparison in surface reconstruction on D-FAUST.

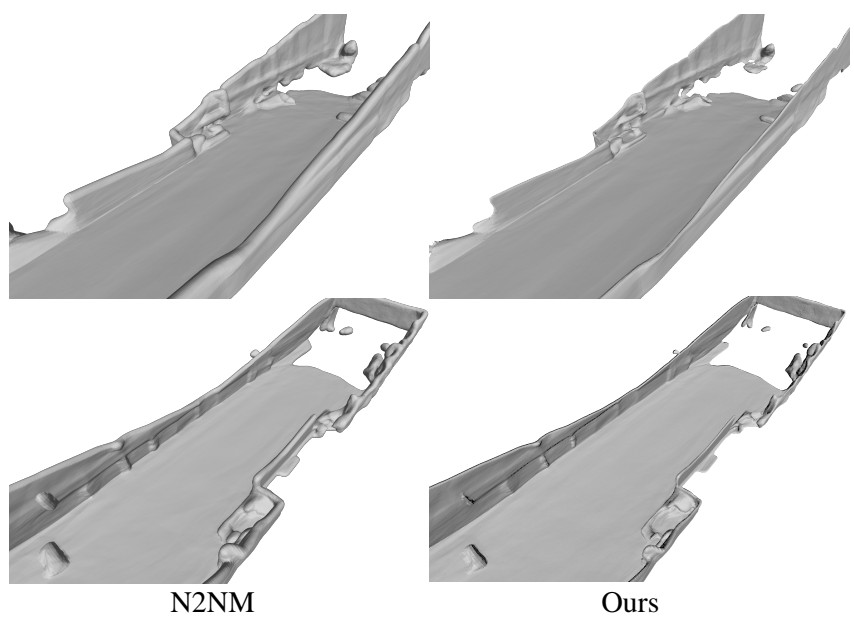

| N2NM | Ours |

Figure 21: Comparison in surface reconstruction on nuScenes.

