# OpenReview forum: "Inferring Neural Signed Distance Functions by Overfitting on Single Noisy Point Clouds through Finetuning Data-Driven based Priors"
_NeurIPS.cc/2024/Conference — NeurIPS 2024 poster_

### Official Review · Reviewer_76DM · 2024-07-04

**Soundness:** 2
**Presentation:** 2
**Contribution:** 2
**Rating:** 4
**Confidence:** 3

**Summary:**

The paper presents a method for learning neural signed distance functions (SDFs) from noisy point clouds. This approach integrates the advantages of data-driven and overfitting-based methods to enhance generalization, accuracy, and inference speed. They employ statistical reasoning within local regions to refine the estimation of SDFs without requiring clean point clouds or signed distance supervision.

**Strengths:**

1. This paper successfully combines data-driven and overfitting-based approaches, leveraging their strengths while mitigating their weaknesses.
2. This paper shows that the method enhances robustness against noise, a common real-world challenge. They effectively use local statistical reasoning to adjust the neural SDFs, improving accuracy in surface reconstruction and denoising tasks, as evidenced by the benchmark results.

**Weaknesses:**

1. There are a lot of typos in the manuscript, e.g., In the Abstract and Introduction, "stat-of-the-art" should be "state-of-the-art"; Line 57: "Learning implicit functions have achieved" should be "Learning implicit functions has achieved"; Line 39: "overfits an neural network" should be " a neural network", "Querys", and so on. In the Abstract, the "prompt" might be "promote". It would be best to check out all the typos.
2. Dependency on initial conditions: The performance might heavily rely on the quality of the initial data-driven priors, which might limit the method's effectiveness when such priors are not well-tuned or applicable.
3. The paper does not extensively discuss the computational demands or scalability when applied to larger datasets (e.g., nuScenes and Waymo) or more complex scenarios.

**Questions:**

1. How does the method perform under limited computational resources? When scaling to larger datasets, is there a significant trade-off between accuracy and computational efficiency?
2. Can the method maintain its performance across different types of noise or corruption that were not part of the training set?
3. How does the method perform when such priors are unavailable, or how sensitive is it to the initial conditions set by these priors?

**Limitations:**

See the "Questions" and "Weaknesses" sections above.

---

> ### Author Rebuttal · Authors · 2024-08-07
>
> 1. Typo
>
> We will correct these typos and proofread the paper more carefully.
>
> 2. Impact on Performance by the Prior
>
> Since we use ground truth signed distances and a mature neural network to learn an implicit function as a prior, the network usually converges quite well. Imperfect priors should not be a concern because of our robust loss. To justify this, we conduct an experiment with a prior that is trained in different iterations, such as 500, 1000, 1500, and 2000 (used in our paper). As shown in Fig. 5 in the rebuttal PDF, we did not see a significant difference between these reconstructions.
>
> 3. Time Complexity
>
> We reported time comparisons in Tab. 9 and Tab. 11. Using a prior can provide a good object-like SDF and adjusts the field accordingly fast. Moreover, our loss that infers information in local regions can also be more efficient than the global strategy, since we avoid points and queries that are far away from each other, please read “G4. Why Local Patches Work Better” above for more details. Thus, our method is much more efficient than globally overfitting based methods. The time is recorded on a single RTX3090 GPU card.
>
> 4. Results on nuScenes
>
> We did report results on complex scenarios like the results on KITTI in Fig. 8 and Paris-rue-Madame in Fig. 9. Both of these results are produced on large scale real scans. Similarly, we additionally report a result on a scene from nuScenes in Fig. 6 in the rebuttal PDF, where our reconstructed mesh surface is nearer to the point and not fat like the N2NM’s result. All these results indicate that our method can handle very large scale and complex real scans.
>
> 5. Running with Limited Resources
>
> Limited computational resources will make a negative impact on the speed but not the accuracy. As we explained in “G4. Why Local Patches Work Better” in the rebuttal above, we train each patch in a batch, which is a good way to run our method on multiple GPU cards in parallel. Another trade-off between accuracy and computational efficiency is to downsample the noisy point cloud. We additionally conducted an experiment with downsampled noisy points in Fig. 2 and Tab. 1 in the rebuttal PDF. We can see that we can work well on much fewer points, and also save some time.
>
> 6. Noise Types
>
> Please read ”G6. Gaussian Noise” in the rebuttal above for more details.
>
> 7. Corruptions
>
> Since our method is targeting on inferring SDF from single noisy shapes, the shapes we used are mainly corrupted by noises, where the prior is learned on clean shapes that are not corrupted at all. But we found our method can generalize the prior on unseen shapes or scenes quite well. Like the objects in SRB or FAMOUS in Fig. 4 and Fig. 5, humans in D-FAUST in Fig. 6 and scenes in 3D scene in Fig. 7, our prior did not see these kinds of data during training. Similarly, our method also did not see any corruptions like some missing structures on real scans like KITTI in Fig. 8, Paris-rue-Madame in Fig.9, and nuScene in Fig. 6 in the rebuttal PDF, but our method also handles these corruptions quite well. This is because our statistical reasoning loss can infer information in local regions in an overfitting manner.
>
> 8. Results without Our Prior
>
> Using data-driven prior is one of our contributions to combine the data-driven based prior with the overfitting based strategy. Our ablation studies in Tab. 9 justify the effect of the prior. Using either random parameters in the network (“Without Prior”) or random initialized shape code (“Without Prior”, “Without Embed”) indicates no prior is used in the experiments, which either lowers the accuracy and slows down the convergence.

---

> ### Comment · Area_Chair_nihP · 2024-08-12
>
> Hi, it's near the end of the rebuttal period. Could you please respond to the rebuttal?

---

> > ### Author Response · Authors · 2024-08-13
> >
> > Dear reviewer 76DM,
> >
> > As the reviewer-author discussion period is about to end, can you please let us know if our rebuttal addressed your concerns? If it is not the case, we are looking forward to taking the last minute to make further explanation and clarification.
> >
> > Thanks,
> >
> > The authors

---

### Official Review · Reviewer_mU7t · 2024-07-09

**Soundness:** 3
**Presentation:** 2
**Contribution:** 2
**Rating:** 5
**Confidence:** 4

**Summary:**

The presented work tackles the problem of reconstructing a shape from a noisy point cloud (PC) into an implicit representation, using the signed distance function (SDF). Recent methods are categorised into i) data-driven approaches that learn a shape prior with a dataset of training shapes, with poor generalization but generally faster, or ii) approaches that overfit on a single PC at a time, with better generalization but slow convergence. The paper proposes a middle ground: finetuning a pre-trained shape prior model to a single PC, using a novel approach that performs "local statistical reasoning": supervising local parts of the shape with multiple sampled noisy PC. The method is experimentally validated against multiple baselines, either data-driven or overfitting-based, and on multiple datasets showing better reconstruction accuracy at lower convergence times. A final ablation study shows that the prior and locality of the loss play a role in the method's performance.

**Strengths:**

1. Experiments are extensive: the PC reconstruction is evaluated against multiple baselines (both data-driven or overfitting-based) on various datasets (ShapeNet, ABC, FAMOUS, SRB, and D-FAUST). In nearly all cases and presented metrics, the proposed approach is the best.
2. I found the locality approach to solving the problem interesting; supervising parts of the shape at a time instead of the full one.
3. Using a shape prior that is unrelated to the target object (such as ShapeNet prior for FAMOUS reconstruction) also works. This suggests that this approach is applicable even if the target shape is from an unknown category (thus helps for generalization).
4. The method is fairly simple and clear, which should make it easily usable.

**Weaknesses:**

1. The main pipeline is fairly simple (fine-tuning a pre-trained model). I believe the locality of the "statistical reasoning" is perhaps the main novelty, and the ablation shows that it improves accuracy and convergence time (§ at L.342), but it is not discussed or explained why using a local approach instead of a global one improves convergence and accuracy.
2. The writing could be improved. More specifically, the experimental results on the different datasets all make more or less the same points and so could be condensed into something more compact. This can free up some space to expand other parts and increase their clarity, e.g., the related works and the "Neural Signed Distance Function." paragraph (L.101).
3. Throughout the work, the noise is assumed (additive) Gaussian. While this is not a weakness in itself, this still is an assumption that should be explained and made clear:
   1. Why is a Gaussian noise model good? Are Lidar data, for instance, usually with gaussian noise and what real use-cases does it apply to? Generally speaking, the motivation of the work could be expanded beyond "reconstructing a PC to an SDF".
   2. "Statistically, the expectation of the zero-level set should have the minimum distance to all the noisy point splitting." (L.131-132) This is true assuming Gaussian noise with expectation 0. With other noise types the loss may not perform as well (e.g., "Text removal" in Noise2Noise [1]).

[1] Lehtinen et al., Noise2Noise: Learning Image Restoration without Clean Data, ICML 2018.

**Questions:**

1. What is the reasoning behind using the locality in the "statistical reasoning" and its better performance?
2. For the ShapeNet dataset, is it per-category and then averaged or trained on multiple categories combined? (and which ones?) Same questions regarding the SRB/FAMOUS experiments with ShapeNet prior.
3. What are the PC sizes? It's hard to appreciate the other parameters without this info. What are the datasets sizes in number of training and test shapes?
4. How does the prior help if it apparently does not need to be from the same dataset as the test shapes (e.g., SRB/FAMOUS experiments)? Is it actually pre-training on any valid SDFs that helps? If so, how do you think it compares to the simple SAL [2] geometric initialization?
5. Why are the metrics (CD-L2/L1, NC, F1-score) not used consistently through the datasets and results?

[2] Atzmon and Lipman, SAL: Sign Agnostic Learning of Shapes from Raw Data, CVPR 2020.

**Limitations:**

Limitations are briefly mentioned in the appendix regarding noise level. What kind of noise level is considered too big? It might also be worth commenting on the noise types (see Weakness 3.), is the method working only for Gaussian noises?

---

> ### Author Rebuttal · Authors · 2024-08-07
>
> 1. Contribution and Novelty
>
> Our novelty is not only the loss but also the way of generalizing the prior. Please see G1-G3 in our rebuttal above.
>
> 2. Why Local Patches Work Better
>
> Please read “G4. Why Local Patches Work Better” in our rebuttal above for the analysis.
>
> 3. Writing
>
> We will follow your advice to polish our writing in our revision.
>
> 4. Gaussian Noise
>
> Please read ”G6. Gaussian Noise” in our rebuttal above for more details.
>
>
> 5. What is Statistical Reasoning and Why Local Patches Work Better
>
> Please read “G5. Statistical Reasoning” and “G4. Why Local Patches Work Better” in our rebuttal above.
>
> 6. Training Data-driven Priors
>
> The prior is learned on multiple shape categories on ShapeNet. We also tried to learn the prior in each category separately, the performance can get even better than we reported in the current submission.
>
> 7. Priors Used on SRB/FAMOUS
>
> As stated in Line 237-239 and Line 253-255, we use the prior learned from ShapeNet, and generalize the learned prior on SRB/FAMOUS for inference. Either SRB or FAMOUS is much different from ShapeNet. Our good performance on these real scans justify the good generalization ability of our method.
>
> 8. Point Cloud Size and Training and Testing Sets
>
> For point clouds for training, we used the data from [50] for fair comparison on ShapeNet, where each point cloud has 3k points. We used over 27k shapes from 13 categories in ShapeNet for learning a data-driven prior, and generalize this prior on SRB, FAMOUS, D-FAUST, 3D scene, KITTI, and Paris-rue-Madame datasets which do not contain a training set. We used nearly 5k shapes from the training set in ABC to learn a prior which is generalized to the testing set, where each shape has 5k to 12k points.
>
>
> For testing shapes, the point number of point clouds is determined by datasets. We used about 7k shapes from 13 categories on ShapeNet, and each point cloud has 3k points. On SRB, there are 5 shapes, and each shape contains about 57k to 95k points with noise. On ABC, the testing set we used include 100 shapes, each shape has 5k to 12k points. On FAMOUS, there are 22 shapes, and each shape has 20k points. On D-FAUST, there are 5 shapes for testing, and each shape has 200k points. On 3D scene, there are 5 scenes, and each scene has about 3900k points. On KITTI, we use one scene which contains 13720k points. And on Paris-rue-Madame, we use one scene which contains 10000k points.
>
> We will make this more clear in our revision.
>
>
> 9. Difference to SAL geometry initialization
>
> Geometry initialization introduced by SAL uses analytically determined parameters to initialize an SDF like a sphere, which requires to set a proper radius and just determines the outside and inside. While ours can produce an object like SDF, which produces more details in the initialization, and more importantly, our prior can adjust the SDF fast according to what it saw during the training.
>
> We do not think any valid SDF can help. Since simple shapes just provide an SDF with a boundary indicating the inside or outside, but can not adjust the SDF with any experience.
>
> We conduct an additional experiment to compare with different initializations in Tab. 2 and Fig. 4 in the rebuttal PDF, including random initialization, geometry initialization, simple square, and ours. We can see our prior can reconstruct more accurate surfaces from single noisy point clouds in much shorter time than any other initializations.
>
> 10. Metrics on Different Benchmarks
>
> For fair comparisons, we follow previous methods to report the results in terms of the same metrics they used, which leads to inconsistent metrics on different benchmarks.
>
> 11. Big Noise Level
>
> There is no standard to define what noise level is a big noise level. But we can handle pretty large noises like what we show in Fig. 1 in the rebuttal PDF. Noises with a variance of 5% are usually used as a large noise by previous methods, but we can handle noises with a 7% variance.

---

> > ### Comment · Reviewer_mU7t · 2024-08-09
> >
> > I thank the authors for their answers and clarifications to my questions, and for the added results in the rebuttal PDF.
> >
> > I believe my weakness #3 regarding Gaussian noise has been mostly answered in the common rebuttal above, with the additional experiment in the rebuttal PDF. Additionally, I had misunderstood that all experiments were using additive Gaussian noise, while some are:
> > > real scans with unknown noise types, such as our results on SRB in Tab.4 and Fig.4, D-FAUST in Tab.6 and Fig.6, 3D scene in Tab.7 and Fig.7, KITTI in Fig.8, and Paris-rue-Madame in Fig.9.
> >
> > Please, see my other comment in the common rebuttal for additional points.

---

> > > ### Author Response · Authors · 2024-08-09
> > > **We are glad to know our rebuttal addressed your concerns**
> > >
> > > Thanks for letting us know our rebuttal addressed your concerns.
> > >
> > > We will response to your comments in the rebuttal above.
> > >
> > > Best,
> > >
> > > The authors

---

### Official Review · Reviewer_fwt4 · 2024-07-09

**Soundness:** 3
**Presentation:** 2
**Contribution:** 2
**Rating:** 5
**Confidence:** 4

**Summary:**

The authors propose an implicit surface reconstruction method from point clouds that uses a learned prior to initialize the optimization of a neural SDF. First, a neural SDF generator based on DeepSDF [66] is trained on a dataset of shapes to learn a prior over shapes. Given a point cloud, both the shape code used as input for the generator, as well as the parameters of the neural SDF are then optimized to fit the point cloud, using an approach based on Noise to Noise Mapping (N2NM) [50], but focusing on local patches instead of the full global point cloud. The authors show that this approach performs better than N2NM as well as several state-of-the-art methods in an extensive evaluation on several datasets.

**Strengths:**

- Using a learned prior as initialization for an optimization-based surface reconstruction method like N2NM has not been done before as far as I can tell.
- Using local patches also seems like a (smaller) contribution to me (the authors show that this contributes to the performance, but it is not very clear to me at this point why it improves performance significantly - this could be improved).
- Results show consistently better reconstruction accuracy than the state-of-the-art.
- The evaluation is extensive and mostly well-done, and convincingly shows and advantage over the state-of-the-art.

**Weaknesses:**

- While the idea of using a learned prior as initialization of an optimization based method has not been done for surface reconstruction as far as I can tell, it is not very surprising that this can be done and is also relatively straight-forward to do. Both stages are essentially the same as existing work [66] and [50], with some deviation from [50] in using local patches. So all-in-all, the technical contribution does not seem huge, although possibly still large enough for acceptance.
- While local patches do seem to improve performance, it is not fully clear why this is the case, and the exact details of using local patches are missing some details in the exposition, making it unclear exactly how they are implemented.
- A few recent methods are missing both from the related work and the comparisons (see below).
- The exposition could use some additional details and clarifications in some parts (see below).

Most of the weaknesses, apart from the first one, can be addressed with text changes to some extent. Given the good performance of the method and the extensive evaluation, I lean towards acceptance.

Details:
- The field of surface reconstruction from point clouds is quite vast, so the authors missed a few works:
  - Neural-Singular-Hessian: Implicit Neural Representation of Unoriented Point Clouds by Enforcing Singular Hessian, Wang et al., TOG 2023
  - Iterative Poisson Surface Reconstruction (iPSR) for Unoriented Points, Hou et al., TOG 2022
  - 3DShape2Vecset: A 3D Shape Representation for Neural Fields and Generative Diffusion Models, Zhang et al., TOG 2023
  - Geoudf: Surface reconstruction from 3d point clouds via geometry-guided distance representation, Ren et al., TOG 2023

- Given good normals, surface reconstruction becomes much easier - the normal computation could be followed by Poisson reconstruction based on the normals for example. Therefore papers to compute oriented normals are relevant, such as:
  - Orienting Point Clouds with Dipole Propagation, Metzer et al., TOG 2021
  - Globally Consistent Normal Orientation for Point Clouds by Regularizing the Winding-Number Field, Xu et al., TOG 2023
  - SHS-Net: Learning Signed Hyper Surfaces for Oriented Normal Estimation of Point Clouds, Li et al., CVPR 2023

- The exposition could use some additional details and clarifications:
  - The approach for using local patches is missing some details: How many patches are used during optimization? In which order are patches evaluated? One after the other? Or is a random patch chosen in each iteration? This should be clarified.
  - The difference of the second stage to [50] could use some more discussion. The authors describe the application of [50] to local patches as main difference (as shown in Eq. 3). But since the expectancy in Eq. 3 is taken over random samples near random patches, how is this different than taking the expectancy over random samples near the whole shape (as in [50])? This could use some  discussion. (Or is the SDF actually optimized separately per patch? From reading I did not get this impression.)
  - [50] also uses a geometric consistency constraint as regularization, although such a regularization is not described by the authors. If it is not used, a discussion why the authors removed it might be useful.
  - c seems to be initialized to a fixed constant vector: the average of all embeddings in the training set, while the original DeepSDF paper uses a c randomly sampled from a Gaussian. Why did the authors use a different initialization for c here? Did the average empirically perform better? This could be discussed.
  - In the paragraph starting at line 129, It should be stated clearly that the method described here is the method proposed in [50] (although evaluated at random samples near random local patches, rather than random samples near the whole shape).

- The evaluation is quite extensive, but would benefit from a few improvements:
  - In the evaluation, it would be good to hear more about how the time used to fine-tune each optimization-based method was chosen. Was each result optimized until convergence? Or otherwise, how was the number of iterations/optimization time chosen?
  - It would be good to add DeepSDF to the comparison, as it is the closest to the initialization used for the optimization stage, and would show how much the fine-tuning  improves accuracy.
  - In the ablation, using a large local region seems worse than using a global approach without local regions. This should be discussed.

- References [52] and [53] are duplicates

**Questions:**

- A clarification of the unclear details regarding local patches would help.
- A clarification of how timings for methods were chosen would help as well.

**Limitations:**

The authors have not discussed clear limitations unfortunately.

---

> ### Author Rebuttal · Authors · 2024-08-07
>
> 1. Contribution and Novelty
>
> Our method is not a simple combination of DeepSDF and noise2noise. Please see G1-G3 in our rebuttal above.
>
> 2. Why Local Patches Work Better
>
> Please read “G4. Why Local Patches Work Better” in our rebuttal above for the analysis.
>
> 3. We will add references you mentioned
>
> We will add these references. The main difference between these methods and ours lies in the ability of reasoning on noisy point clouds in an overfitting manner. This ability significantly improves the generalization performance on unseen noisy point clouds.
>
> 4. Local Patches in Optimization
>
> We randomly sample a noisy patch and a set of queries within the patch in each iteration. Thus, there is no order needed. We use the loss difference between two successive iterations as a metric to determine the convergence, which can be used to conduct time comparisons in ablation studies. The optimization is usually converged in 4000 iterations, and no more than 4000 patches are used. We will make these details more clear in our revision.
>
> 5. Difference to the Global Mapping [50]
>
> Please read “G4. Why Local Patches Work Better” in our rebuttal above before going ahead.
>
> Besides the inaccuracy and inefficiency in inference, another downside of [50] is the requirement of additional constraints on the pulling, i.e., each travel distance should be as small as possible, which makes sure the learned distance is the minimum to a surface, as shown in Fig. 3 in the original paper of [50]. The reason why [50] needs a constraint like this is two fold. One is that the optimization of SDF is still not converged well during inference, and it produces lots of uncertainty, thus queries can not get pulled at the right place. The other is that the loss of noise2noise just pushes the pulled queries to be as near to the noisy patch as possible, but does not constrain how the pulling will be conducted. Thus, our local patches resolve this issue with a patch limit, which also gets rid of the regularization term. Our preliminary results showed that the regularization term does not improve the performance but slows down the optimization a lot. We will make this more clear in our revision.
>
> 6. Shape Embedding c
>
> Please read “G3. Traditional test-time optimization vs ours” in our rebuttal above before going ahead.
>
> Using averaged shape code as an initialization is a key point to generalize our prior with an overfitting loss.  As shown in our ablation studies in Tab. 9, using randomly initialized shape code (“Without Embed”) makes a negative impact on the accuracy and efficiency. Our averaged shape code takes a good advantage of the prior space, and provides an object-like SDF to start the inference with uncertainty of SDF using the overfitting loss.
>
> 7. Convergence Metric
>
> We use the loss difference between two successive iterations as a metric to determine the convergence, as shown in Tab. 9 and Tab. 11. The optimization is usually converged in 4000 iterations, as stated in Line 164. Each result was recorded when the optimization converges. We will make these details more clear in our revision.
>
> 8. Comparisons with DeepSDF
>
> We compare it with DeepSDF in Tab. 9. As stated in Line 329-330 and the result (“Fixed Param”) in ablation studies in Tab. 9, the auto-decoding introduced in DeepSDF can not work well with the overfitting loss on single noisy point clouds. In almost all cases, it does not reconstruct a plausible shape. As analyzed in “G3. Traditional test-time optimization vs ours”, the signed distances at the same locations inferred by the overfitting loss from a single noisy point cloud are fluctuating iteration by iteration, which is not like the GT signed distances, i.e., constant values, used in DeepSDF. This makes the prior generalization different a lot.
>
> 9. Why Larger Patches Do Not Work Well
>
> Please read “G4. Why Local Patches Work Better” in our rebuttal above going ahead.
>
> With larger patches, more queries and noisy points that are far away from each other are paired to infer the SDF, which is quite similar to the global strategy [50].  This reduces the reconstruction accuracy and efficiency.

---

> > ### Comment · Reviewer_fwt4 · 2024-08-09
> >
> > Thanks to the authors for the detailed replies.
> >
> > Regarding the contributions and novelty, what I meant with the paper using the learned prior as initialization is not that the optimization is like in DeepSDF, but that (apart from the local patches), the optimization is very similar to Noise-to-Noise, but using DeepSDF as initialization for the neural SDF that is being fitted during the optimization. While the choice of initial shape code c is interesting, this choice is not ablated as far as I can tell (adding an ablation would be good actually, especially if it is claimed as contribution), and also does not add a very large contribution by itself. Overall though, given the good performance, I still think the contribution is probably large enough for acceptance.
> >
> > Regarding the discussion why local patches improve performance, I think the discussion provided in the rebuttal seems reasonable to some extent. If point cloud and query densities do not match across the shape, then queries may be matched to more distant points, and the local patches essentially stratify the problem by confining it to local patches. At this point though - why not use the Chamfer Distance instead? The CD would not have this problem with different point and query densities. Is it because we still want the behavior of EMD with its longer distance matches rather than CD in local patches? So overall, while this explanation is already reasonable if clarified a bit, the discussion of this point could also go a bit more in depth, especially since this is one of the main claimed contributions. The discussion of this point (whether it is a clarified version of the current discussion, or a more in-depth discussion) should be added to the paper.
> >
> > Regarding the discussion why using an average shape code for c is better than a random shape code, the only relevant information I can find in the response (apart from references to the ablation) is that the average shape code gives a more 'object-like' appearance. This could some more elaboration. What does 'object-like' mean? Is the average shape code c typically closer to the optimum? Why is this the case? Maybe because the distribution of shape codes used by typical shapes is different from a Gaussian distribution and the average is better centered in this actual distribution of shape codes, so the initial shape output by the neural SDF is already closer to the target shape compared to a random shape code? A somewhat more elaborate discussion of this point would improve the motivation of this design choice.

---

> > > ### Author Response · Authors · 2024-08-09
> > >
> > > **1. Ablation Studies on Shape Codes**
> > >
> > > We did two ablation studies related to shape codes. One is the code length in Tab.8, the other is the code initialization in Tab.9. In Tab.9, we tried random initialization for shape code in “Without Prior” (using network parameters that are learnable and randomly initialized), “Without Embed” (using network parameters that are learnable and initialized with our prior), and “Fixed Param” (using network parameters that are not learnable and initialized with our prior).
> > >
> > > Moreover, as we mentioned in c) in “G3. Traditional test-time optimization vs ours” in the rebuttal above, we also tried various normalizations based on the random initialization, which produces too bad results to get reported in the paper. In addition, we also tried some other variations like our averaged code in our preliminary results, such as randomly selecting one shape code from the training set as the initialization. Similarly, randomly selecting also produces too bad results to get reported in the paper.
> > >
> > > We will follow your advice to reorganize our ablation study on shape codes in our revision. Please also let us know if you would like to see any other alternatives on code initialization.
> > >
> > > **2. How about CD?**
> > >
> > > You raised a great point. Actually, we followed [50] to use EMD to report our results. As a main contribution, the authors of [50] did a pretty good job to find that EMD works while CD does not work, please see Fig.4 and ablation studies “CD” in Tab.11 in their paper for more. But we are different, and we make everything happen in a patch rather than over the whole shape, which changes the results with CD a little bit. In our preliminary results, we found that the CD can produce much better results in our method than in [50], but the results are still much worse than the ones with EMD. Another consideration of using EMD is for fair comparison with [50].
> > >
> > > Although CD can handle density difference between noises and queries and produces more reasonable results in local patches, it does not have one thing that is the key for the success as EMD, which is the one to one correspondence. CD involves many to one correspondence which is not efficient and targeted in statistical noise to noise mapping.
> > >
> > > We will follow your advice and add a discussion on this in our revision.
> > >
> > > **3. Why does averaged shape code work?**
> > >
> > > The “Object-like” SDF is basically a shape, since the code is averaged over all training shapes. With the pretrained network parameters, that is our prior, at the very beginning, this averaged shape code can produce a shape that may be very similar to some shapes in the training set, which however is not necessarily similar to the noisy point cloud. This means that it is not nearer to the optimal point or closer to the target shape, but provides a relatively stable area for the optimization.
> > >
> > > The reason why we use averaged shape code is three fold.
> > >
> > > i) One is that it can not only initialize an SDF with inside and outside, which is basically the same purpose as the geometry initialization (initializes network parameters so that a sphere like SDF is produced) introduced in SAL, but also inherit some geometry details from the training dataset.
> > >
> > > ii) Another one is that it provides a great scope for the optimization to search for a result. This is because the area where the averaged shape code is gathers more shape codes that represent meaningful shapes in the training set than other areas in the shape code space. This area provides a large enough candidate pool for our inference loss, which avoids the optimization to fly away due to the large change of shape code updated by the inferred information, such as signed distances in our case, which is fluctuating iteration by iteration, even at the same locations.
> > >
> > > iii) Based on ii), the network parameters which saw various shapes during training can work with the shape code to fast update the SDF according to the inferred information from the loss.
> > >
> > > These three folds make our averaged shape code is much better than the random initialization which provides a start where there may be no other learned shape codes at all, which makes it hard to find optimization directions in the following.
> > >
> > > We will follow your advice and add this discussion in our revision.

---

### Official Review · Reviewer_AdmT · 2024-07-13

**Soundness:** 2
**Presentation:** 2
**Contribution:** 3
**Rating:** 4
**Confidence:** 4

**Summary:**

The paper proposes 3D shape reconstruction given a noisy point cloud using a test-time optimization approach. The paper proposes a new loss function that can work directly on noisy point clouds without the need of ground truth normals or SDF. The approach involves learning a network (DeepSDF) to predict SDFs and later fine-tune the network on a single shape during test. The learned priors reduce the test-time fitting and leads to better reconstruction results.

**Strengths:**

1. The paper proposes a simple and easy to implement approach to 3D reconstruction that works on a bit noisy point clouds.
2. The better quality results and fast inference has a broad significance in 3D reconstruction.

**Weaknesses:**

1. In the introduction, the test-time optimization is hailed as a novel thing, however previous methods (including DeepSDF) has already proposed it.
2. The main contribution is the loss function defined using "pulling" operation, which forces the surface defined by the projected points close to the input points be similar to the input point clouds. However, why this loss function should work in a noisy point cloud setting is not well motivated and well explained.
3. The writing of the method section can be significantly improved. For example, the symbol $f$ seems to represent both the neural network that predicts the signed distance field and the field it self. More about it in the Questions section.
4. The use of the phrase "statistical reasoning"is quite vague. The phrase is used throughout the manuscript but does not convey anything meaningful.
5. It is not clear whether the baselines in the experiment section are also trained on noisy data to make them robust.
6. Comparison with Points2Surf [21] method is missing. I think it work quite well on shapes in Figure 4.
7. What does "extreme noise" mean in the Table 13? What does it translate to in percentage of the scale of the scene.
8. How do you achieve 100% sparsity in Table 14? Does it imply no points?
9. Noise level of 0.5% is quite low to claim that the method is robust.

**Questions:**

1. Please improve the notations in the method section. The vectors should be bold. Use different symbols for parameterized function and output.
2. The motivation of the loss function is not clear. Perhaps a diagram would enhance the understanding for a reader.

**Limitations:**

1. Limitations regarding pose variations and incomplete data (holes) are not discussed.

---

> ### Author Rebuttal · Authors · 2024-08-07
>
> 1. Why pulling works
>
> Pulling queries towards surface points has been widely used to estimate a signed distance function from a point cloud. We can use a neural network to infer the SDF by pulling randomly sampled queries to their nearest surface points using the predicted signed distances and gradients. We train this network by minimizing the distance between the pulled queries and queries’ nearest surface points.
>
> However, it is not a good way to infer an SDF through pulling queries on a noisy point cloud, since queries’ nearest surface points are not accurate to find due to noise corruption. To resolve this issue, we got inspiration from [1] and [2], which infers the clean 2D and 3D information from noisy observation respectively. Our statistical reasoning in a local region managed to estimate an SDF by minimizing the distance between different sets of pulled queries that are randomly sampled around and their commonly shared noisy patch. The rationale behind this is that the different sets of pulled queries are forced to be as near to the same noisy point patch as possible, when we do this over different noisy point patches that overlap with each other, the optimization converges to a consistent zero-level set.
>
> [1] Noise2noise: Learning Image Restoration without Clean Data
>
> [2] Learning Signed Distance Functions from Noisy 3D Point Clouds via Noise to Noise Mapping
>
> 2. Notation
>
> We will revise the notation to make them more specific and clear.
>
> 3. Statistical Reasoning
>
> Please read “G5. Statistical Reasoning” in the rebuttal above.
>
> 4. Baselines
>
> Data-driven based methods such as IMLS and ConOcc were pre-trained on point clouds with noise of different variances, so that they can work with various noises during inference. While overfitting based methods directly learn SDFs on single shapes in the testing set. Both data-driven based and overfitting based methods use the same set of noisy point clouds as ours for evaluations.
>
> 5. Comparison with Points2Surf
>
> We compared the data-driven based method Points2Surf on ABC in Tab. 3 (“P2S”), Fig. 3 (“P2S”), and Fig. 12 (“P2S”). Numerical and visual comparisons indicate that our method significantly outperforms Points2Surf. Since Points2Surf is a relatively old method that was proposed back in 2020, and its following methods like PCP [3] and OnSurf [4] have reported much better performance on FAMOUS dataset, we just list PCP and OnSurf as baselines in Tab. 5. Moreover, Points2Surf did not report its results on SRB in the original paper, and we failed to reproduce plausible results on SRB by running its code as well, because of its poor generalization ability on SRB. Thus, we did not compare Points2Surf in Fig.4.
>
> [3] Surface reconstruction from point clouds by learning predictive context priors
> [4] Reconstructing surfaces for sparse point clouds with on-surface priors
>
> 6. Extreme Noise in Tab. 13
>
> The noise levels of middle and max come from the ABC dataset released by Points2Surf. The middle indicates noises with a variance of 0.01L, where L is the longest edge of the bounding box. The max indicates noises with a variance of 0.05L. Our extreme noise comes with a variance of 0.07L.
>
>
> 7. 100\% sparsity in Tab. 14
>
> As stated in Line 361-362, the percentages shown in Tab. 14 are ratios of left over points after the downsampling. Thus, 100\% indicates the results without downsampling. We will clarify this in the revision.
>
> 8. 0.5\% noise variance
>
> We use 0.5\% noise variance on ShapeNet for fair comparisons with previous methods. As shown on ABC in Tab. 3, FAMOUS in Tab. 5, shapes with extreme noise in Tab. 13, and the additional results in Fig. 1 in the rebuttal PDF, our method can handle point clouds with noise variance as large as 7\%.
>
> 9. Understanding of Our Loss
>
> The diagram of our loss is shown as L_{Local} on the right in Fig. 1. For a noisy point patch, we randomly sample a set of queries within the noisy point patch. Then, we use the learned SDF to pull them onto the zero-level set, and we minimize the EMD distance between the set of pulled queries and the noisy point patch.

---

> ### Comment · Area_Chair_nihP · 2024-08-12
>
> Hi, could you please chime in? It's near the end of the discussion period!

---

> > ### Author Response · Authors · 2024-08-13
> >
> > Dear reviewer AdmT,
> >
> > As the reviewer-author discussion period is about to end, can you please let us know if our rebuttal addressed your concerns? If it is not the case, we are looking forward to taking the last minute to make further explanation and clarification.
> >
> > Thanks,
> >
> > The authors

---

### Author Rebuttal · Authors · 2024-08-07

We appreciate the reviewers' valuable comments, which highlighted our simple and interesting method (AdmT, mU7t), strong performance and extensive experiments (fwt4, mU7t, 76DM), and the broad significance and usefulness of our work (mU7t, AdmT).

G1. Our Novelty

Our novelty lies in how to combine data-driven based priors with the overfitting based strategy but rather the test-time optimization. Our framework is much different from the test-time optimization introduced in DeepSDF, since DeepSDF uses ground truth (GT) signed distances during both training and testing, which is relatively easy to generalize the prior learned during training. While we use different kinds of supervision during training and testing, which brings challenges for prior generalization. Specifically, during training, we use GT signed distances to learn what shapes look like as a prior, while using a loss to infer plausible signed distances from noisy point clouds fast based on the prior during testing. The signed distances inferred in iterative optimization may be slightly fluctuating iteration by iteration,  which brings significant uncertainty and makes it hard to optimize the code with fixed parameters like auto-decoder, while GT signed distances are constant. Thus, this challenge requires a novel way of generalizing a prior in an overfitting strategy.

G2. Our Contribution

Our main contributions are two folds. One is the noise to noise mapping in local regions, the other is how to leverage the prior as an initialization for inferring a SDF from a single noisy point cloud. Our first contribution leads to a loss that can infer more accurate and sharper geometry from a single noisy point cloud. The second contribution leads to a novel way of using the prior, which is the key to the fast convergence and good generalization of the prior. This way is not a simple variation of DeepSDF and noise2noise, please read “G3. Traditional test-time optimization vs ours” for more details.

G3. Traditional test-time optimization vs ours

The test-time optimization introduced in DeepSDF just optimizes a learnable code with fixed network parameters (a prior). It works well if the supervision used during testing is the same kind as the one used during training, such as GT signed distances. The difference to ours lies in that we do not access GT signed distances during testing, but just noisy point clouds, so we leverage a loss to infer signed distances. The inferred supervision is different from GT signed distances, since ground truth signed distances are constant values which keeps exactly the same in different iterations, while the signed distances inferred by pulling are fluctuating iteration by iteration, even at the same locations, which makes the learned prior hard to generalize according to the inferred supervision. We tried the following options at very beginning, but none of them can make the prior work with an overfitting based loss.

a)Randomly code + fixed network parameters ( DeepSDF)

b)Randomly code + learnable network parameters (prior initialization)

c)Randomly code with normalization + learnable network parameters (prior initialization)

Ours shown below work well with an overfitting based loss quite well.

d)averaged shape code + learnable network parameters (prior initialization)

This initialization provides us a good object-like SDF that is robust enough to the information fluctuation inferred by the overfitting loss. Moreover, the prior can adjust the SDF fast.

G4. Why Local Patches Work Better

As stated in Line 122-132, we randomly sample a point patch from a noisy point cloud, and a set of queries within this patch, and pair these two sets with each other to learn a SDF. Different from the global method which samples both noisy points and queries all over a shape, we have a clear boundary each time, which makes the SDF inference more targeted and efficient. This is because we avoid noisy points and queries that are too far away from each other, which makes no sense to pair them together for inference. Our ablation studies in Tab. 11 justify that our local strategy can significantly improve the reconstruction accuracy and reduce the inference time.

G5. Statistical Reasoning

As stated in Lines 129-132, we aim to reason a neural SDF f from a noisy point cloud. Since we randomly sample queries and noisy points, we statistically minimize the expectation of the distance between different sets of pulled queries and noisy points. Statistically, the expectation of the zero-level set should have the minimum distance to all the noisy points. We regard this inferring process from a noisy point cloud as statistical reasoning. We will make this more clear in our revision.

G6. Gaussian Noise

Current studies have shown that the distribution of real noises from scanning sensors is very close to Gaussian distribution, please see Fig.7 in [1]. Vast amounts of literatures on point clouds also commonly use Gaussian distribution to add noises. Although our optimization minimizes the expectation, our method can work beyond Gaussian noises.  We also perform quite well on real scans with unknown noise types, such as our results on SRB in Tab.4 and Fig.4, D-FAUST in Tab.6 and Fig.6, 3D scene in Tab.7 and Fig.7, KITTI in Fig.8, and Paris-rue-Madame in Fig.9.

We additionally conduct an experiment to show our performance with various noise types, i.e., impulse noise, quantization noise, Laplacian noise, and Gaussian noise. Visual comparison in Fig. 3 in the rebuttal PDF justifies that we can also handle other types of noise quite well. Moreover, we also tried more challenging cases with nonuniform noises which do not have a zero expectation across a shape, like a shape with only a half point having noises or a shape with several patches having noises. The result in Fig. 7 in the rebuttal PDF shows that our method can also handle nonuniform noises well.

[1] Noise characterization of depth sensors for surface inspections

---

> ### Comment · Reviewer_mU7t · 2024-08-09
>
> **Local patches work better**
> > This is because we avoid noisy points and queries that are too far away from each other,
>
> From my understanding, this is affected by the point cloud density: sparser PC might have fewer noisy points where the queries are projected. If this is correct, then using a local patch against the global shape should not differ much in that regard, *if* the number of sampled points and queries $U$ is scaled appropriately (ignoring memory restriction for the sake of the discussion). Is the local patch approach leveraging something else?
>
> In the “Global and Local.” ablation (L.342), how are sampled the queries and noisy points for the global approach?
>
> **DeepSDF test-time optimization.**
>
> Note that DeepSDF has also been shown to work to reconstruct partial and noisy point clouds in its work, see Section 6.3 and Fig. 8 and 9 in their work. It does however require some pseudo-SDF to be constructed from the PC.

---

> > ### Author Response · Authors · 2024-08-09
> >
> > Thanks for your comments. Please find our explanation and clarification below.
> >
> > **1. Too far in “we avoid noisy points and queries that are far away from each other” is not related to sparsity.**
> >
> > When we say too far, that is really far, and not caused by sparsity. Taking humans in D-FAUST dataset for example, if we sample some queries near the human’s head, the queries may get paired with noisy points on the human’s feet in the global reasoning method, due to the total randomness over the shape. Obviously, this kind of paires should be avoided since the statistical reasoning on these pairs does not contribute to the learning of SDFs, in contrast, it wastes some time and adds some additional burden on optimization. To relieve this burden, the optimization also requires some kinds of regularization, like the one in [50], to constrain the movement in the pulling, so that the field is inferred as an signed distance field. Please read more about the regularization in our reply to reviewer fwt4’s 5th question.
> >
> > We establish queries and noisy points pairs in local patches, which filters out unwanted pairs and does not need additional regularization or other things.
> >
> > **2. Ablation on Global and Local in L342**
> >
> > The experimental setting is the same except how we establish noisy points and queries. For the “Global”, we keep the sampling and pairing strategies the same as [50], which randomly samples a set of queries and a set of noisy points without the patch limits. The unwanted pairs in “Global” degenerates the accuracy and slows down the convergence.
> >
> > **3. Difference to DeepSDF test-time optimization in Fig.8 and Fig.9**
> >
> > i) We can infer signed distances but DeepSDF can not infer but using inferred or GT signed distances
> >
> > DeepSDF requires GT or inferred signed distances to conduct the test-time optimization. It does not infer signed distances as ours, but just uses the ones that are ready to use. As stated in the second paragraph in the left column on Page 7 in the DeepSDF paper, they generate SDF point samples using depth maps and normals to sample points with known distances, which establish the GT signed distances. Then, DeepSDF uses maximum a posterior estimation to find the shape code (using a prior represented by fixed network parameters) that best explains a partial shape observation in Fig.8 and Fig.9, using Eq.10 with a loss function in Eq.4.
> >
> > Ours does not require depth images and normals to estimate signed distances, but directly infer signed distances from noisy points. This means if you only have a piece of noisy point cloud, DeepSDF can not conduct the test-time optimization but we can handle it quite well.
> >
> > ii) During test-time optimization, our can handle huge uncertainty while DeepSDF can not
> >
> > Because of i), DeepSDF uses constant signed distance values as supervision, while we use signed distances inferred by the loss which produces fluctuating signed distances iteration by iteration, even at the same location.
> >
> > iii) For prior generalization, we fine-tune the network while DeepSDF uses fixed network parameters
> >
> > DeepSDF uses the same kind of supervision during both training and testing, such as signed distances, which makes prior generalization relatively easy. Thus, DeepSDF just uses fixed network parameters for prior generalization during test-time optimization.
> >
> > However, we usually do not have signed distances for most scenarios during testing, but just raw observations, like noisy points. Thus, we need novel ways to infer signed distances, while turning using fixed parameters for prior generalization problems. Our method fine-tunes the network with a prior initialization for prior generalization.

---

> > > ### Comment · Reviewer_mU7t · 2024-08-12
> > >
> > > Thank you for the added clarifications.
> > > Indeed, EMD may match some queries and points that are far away on the shape because of the one-to-one mapping and inherent noise in the problem. I see now how local patches can help alleviate this issue by only considering a smaller part of the shape at a time.
> > >
> > > I would suggest adding these explanations to the paper to strengthen the use of local patches, if the authors find it relevant.

---

> > > > ### Author Response · Authors · 2024-08-12
> > > >
> > > > Thanks for your comments and advice. We will add this discussion in our revision.
> > > >
> > > >
> > > > Best,
> > > >
> > > > The authors

---

### Author Response · Authors · 2024-08-08
**We value the opportunity to discuss with you**

Dear reviewers,

Thanks for your valuable comments. We will be happy to take any questions. Please let us know if our rebuttal address your concerns.

Thanks,
The authors

---

> ### Author Response · Authors · 2024-08-11
>
> Dear reviewers,
>
> As the reviewer-author discussion period is about to end, we are looking forward to your feedback on our rebuttal. Please let us know if our responses address your concerns. We would be glad to make any further explanation and clarification.
>
> Thanks,
>
> The authors

---

### Decision · Program_Chairs · 2024-09-25

**Decision:**

Accept (poster)

**Comment:**

Initially, the reviewers had mixed responses, but after the rebuttal and the discussions, the majority of the reviewers voted in favour of acceptance. Except for Reviewer 76DM, who never returned after the initial review despite both the authors and the AC trying to reach the reviewer, all reviewers recommended accepting the paper. Reviewer AdmT's rating currently shows up as borderline reject, but in the discussions recommended acceptance, hence this seems to be just that the reviewer did not update the rating after the discussions.

In short, there were concerns about the limited novelty, but reviewers agreed that the claims made in the paper have been adequately verified, and the paper also shows a significant performance gain. The AC agrees with this recommendation and recommends accepting the paper. The authors are, however, strongly encouraged to incorporate the discussions during the rebuttal to the main paper, as it played a pivotal role in the paper's acceptance.